



**Impact of intraseasonal wind bursts on SST variability in the**
**far eastern Tropical Atlantic Ocean during boreal spring**
**2005 and 2006. Focus on the mid-May 2005 event.**
Gaëlle Herbert[1], Bernard Bourlès[1]
[1]:Institut de Recherche pour le Développement (IRD), Laboratoire d'Etudes Géophysiques et Océanographie Spatiale
(LEGOS), Brest, France.
*Correspondence to*: Gaëlle Herbert (gaelle.herbert@ird.fr)
**Abstract.** The impact of spring intraseasonal wind bursts on sea surface temperature variability in the eastern
Tropical Atlantic Ocean in 2005 and 2006 is investigated using numerical simulation and observations. We
specially focus on the few documented coastal region east of 5° E and between the equator and 7° S. For both
years, the southerly winds strengthening induced cooling events through i) upwelling processes; ii) vertical
mixing due to vertical shear of zonal current; and for some particular events iii) a decrease of incoming surface
shortwave radiation. The strength of the cooling events was modulated by subsurface conditions affected by the
arrival of Kelvin waves from the west influencing the depth of the thermocline. Once impinging the eastern
boundary, the Kelvin waves excited westward-propagating Rossby waves which, combined with the effect of
enhanced westward surface currents, contributed to the westward extension of the cold water. A particularly
strong wind event occurred in mid-May 2005 and caused an anomalous strong cooling off Cape-Lopez and in
the whole eastern Tropical Atlantic Ocean. From the analysis of oceanic and atmospheric conditions during this
particular event, it appears that anomalous strong spring wind strengthening associated to anomalous strong
Hadley cell activity made the event as a decisive event which prematurely triggered the rainfall coastal onset in
the northern Gulf of Guinea. Results show that no similar atmospheric conditions were observed over the 1998-
2008 period. It is also found that the anomalous oceanic and atmospheric conditions associated to the event
exerted strong influence on rainfall off Northeast Brazil. This study highlights the different processes through
which the wind power from South Atlantic is brought to the ocean in the Gulf of Guinea and emphasizes the
need to further document and monitor the South Atlantic region.
**1.Introduction**
The eastern equatorial Atlantic Ocean shows a pronounced seasonal cycle in sea surface temperature (SST)
(Wauthy, 1983; Mitchell and Wallace, 1992). One strong signature on the SST seasonal cycle in the eastern
equatorial Atlantic is the Atlantic cold tongue (ACT) (Zebiak, 1993) characterized by a fast drop of SST (up to
7° C) in boreal spring and summer slightly south of the equator and east of 20°W (Merle, 1980; Picaut, 1983).
During boreal summer, the southern boundary of this cooler temperature connects progressively with the austral



winter cooling of the Southern hemisphere SSTs. A number of observational (Merle, 1980; Foltz et al., 2003)
and modeling (Philander and Pacanowski, 1986; Yu et al., 2006; Peter et al., 2006) studies show that the
development of the ACT is driven by the seasonal increase of the Southern Hemisphere trade winds during late
boreal winter to early summer (Brandt et al., 2011) associated to the meridional displacement of the Inter-
Tropical Convergence Zone (ITCZ) (Picaut, 1983; Colin, 1989; Waliser and Gautier, 1993; Nobre and Shukla,
1996). The equatorial cooling would be regulated by a coupling between thermocline shoaling, subsurface
dynamics (Yu et al., 2006; Peter et al., 2006; Wade et al., 2011; Jouanno et al., 2011) including turbulent
mixing, vertical advection and entrainment, as well as horizontal advection. The equatorial thermocline shoaling
is the consequence of local and remote wind forcing: the strengthening of easterly winds in the western
equatorial Atlantic remotely forces the seasonal upwelling in the eastern part of the basin via equatorial Kelvin
waves (Moore et al., 1978; Adamec and O'Brien, 1978; Busalacchi and Picaut, 1983; McCreary et al., 1984).
Besides the dominant seasonal cycle, the eastern tropical Atlantic is under the influence of meridional southerly
winds (Picaut, 1984) which fluctuate with a period close to 15 days (Krishnamurti, 1980; de Coëtlogon et al.,
2010; Jouanno et al., 2013). These intraseasonal wind fluctuations are therefore expected to be a major
contributor to the seasonal SST cooling and their fluctuations occur as a vector of energy and momentum from
the South Atlantic to the eastern equatorial Atlantic. A connection between the strength of the St. Helena
Anticyclone and SST anomalies in the southeastern tropical Atlantic has been described by Lübbecke et al.
(2014). These authors suggest that the St. Helena Anticyclone variability might be an importance source of
anomalous tropical Atlantic wind power which affects SST in the eastern equatorial Atlantic via several
mechanisms: zonal wind stress changes in the western equatorial basin, wave adjustment, meridional advection
of subsurface temperature anomalies, intraseasonal wind stress variations, and possibly even other mechanisms.
Through the in situ data analysis of AMMA/EGEE cruises (Redelsperger et al., 2006; Bourlès et al., 2007)
carried out in 2005 and 2006, Marin et al. (2009) show that the SST seasonal cooling at the equator east of 10°
W is not smooth but results from the succession of short-duration cooling events generated by southeasterly
wind bursts due to the fluctuating St. Helena Anticyclone. In addition, according to Leduc-Leballeur et al.
(2013), the sharp and durable change in the atmospheric circulation in the northern Gulf of Guinea (durably
strong southerlies north of equator) takes place through an abrupt seasonal transition prepared by a succession of
southerly wind bursts and possibly triggered by a significantly stronger wind burst. The southerly wind bursts
occurring in spring in the Gulf of Guinea thus would play an important role in driving precipitation pattern in
the area through air-sea interactions (de Coëtlogon et al., 2010; Nicholson and Dezfuli, 2013) and coupling
between the ACT and the West Africa Monsoon (WAM).
Improving our understanding of the impact of such wind bursts on SST variability at intraseasonal scale in the
eastern Tropical Atlantic is important through its link with the regional climate. However, while the ACT and
Angola-Benguela regions have been the object of many studies, the dynamics and SST variability of the coastal
eastern region is much less documented.





In this study, we therefore first focus our analysis off Cape-Lopez (defined from 0° N-7° S; 5° E-14° E and
hereafter called CLR for 'Cape-Lopez region') and aim to improve understanding of its seasonal SST variability
and the impact of intraseasonal winds on SST variability during spring and summer. To this end, we use
regional high resolution model results as well as satellite SST data and sea surface height observations. We first
use model outputs from 1998 to 2008 to analyze the seasonal cycle in CLR and to highlight its interannual
variability, and then we specially focus on the years 2005 and 2006 to investigate the SST response of
intraseasonal wind forcing. These two particular years were largely investigated during the African Monsoon
Multidisciplinary Analyses (AMMA) experiment (Redelsperger et al., 2006). The year 2005 is characterized by
the lowest SST values in the ACT during the past 3 decades (along with 1982), while 2006 is considered as a
normal year (Caniaux et al., 2011). Also, 2005 exhibits the earliest development of the ACT. The study of SST
variability at intraseasonal scale during these two years is thus interesting for better understanding their
observed differences in SST seasonal conditions. These two particular years have been also chosen by Marin et
al. (2009) to study the variability of the properties of the ACT. Their study concerned the equatorial area west of
4° E, whereas we propose to focus in CLR, east of 5° E where coastal processes are expected to be involved.
Most studies on CLR focused on the analysis of observational dataset to examine the hydrology and its seasonal
variation along the frontal (coastal) region of Congo (e.g. Merle, 1972; Piton, 1988) or on the impact of Congo
River on SST and mixed layer (e.g. Materia et al., 2012; Denamiel et al., 2013; White and Toumi, 2014) but, to
our knowledge, no detailed analysis of SST variability at seasonal and intraseasonal time scales have been
realized. A better understanding of ocean-atmosphere interactions in this region is thus needed. Some previous
studies related to the whole eastern Tropical Atlantic (Gulf of Guinea) suggest that multiplicity of processes
could be in play in CLR, coupling remote and local forcing, and combined with the very low thermal inertia of
the mixed layer depth. For example, Giordani et al. (2013) show from regional model results that horizontal
advection, entrainment, and turbulent mixing significantly contribute to the heat budget east of 3°W because of
the very thin mixed layer. The upper layers of the north CLR might also be impacted by vertical mixing induced
by the intense current vertical shear between the South Equatorial Current, flowing westward at the surface, and
the subsurface eastward Equatorial Under-Current. In addition to local forcing, the area is also under the
influence of the arrival of equatorial Kelvin waves from West and their reflection, once reaching the African
coast, poleward as coastally trapped waves and westward as Rossby waves (Moore, 1968; McCreary, 1976;
Moore and Philander, 1977). The principal source of the equatorial Kelvin waves has been usually related to the
western equatorial zonal wind changes during late boreal winter to early summer (e.g.; Philander, 1990). In
order to better understand the triggered mechanism of Kelvin waves generation which conditions the mixed
layer properties in the CLR, another purpose of this study is thus to identify the atmospheric conditions
coinciding with the Kelvin waves generation in the West of the basin during winter 2005 and 2006.
Several studies (e.g. Okumura and Xie, 2004; Caniaux et al., 2011; Nguyen et al., 2011; Thorncroft et al., 2011)
put into evidence a high correlation between the ACT and the WAM onset in the Sahelian region. Based on an
analysis of 27 years of data, Caniaux et al. (2011) identified the year 2005 as the year with the earliest WAM





onset date (around 19 May 2005 whereas they define the mean onset date on 23 June +/-8 days). According to
Marin et al. (2009), the time shift in the development of the ACT between 2005 and 2006 is related to a
particular wind burst event in mid-May 2005. This mid-May 2005 event therefore appears as exerting a strong
influence on the WAM. In a second part of the study, we thus focus on this particular wind event that preceded a
strong cold event in the far eastern Tropical Atlantic along with an early ACT development. We aim to describe
i) the atmospheric and oceanic conditions during this particular event; ii) to what extent it is involved in the
WAM system; and iii) which processes make it an exceptional event.

The remainder of the paper is organized as follows. In Sect. 2, the model and observational data used in this
study are described. The seasonal and interannual variability of SST, winds, currents, 20° C-isotherm depth and
sea surface heat flux in the CLR are analyzed in Sect. 3. The cooling events generated in response to southerly
wind bursts and the other forcing mechanisms implied in the CLR are investigated in details for the years 2005
and 2006 in Sect. 4. In Sect. 5, we focus our analysis on the unusual wind burst occurring in mid-May 2005.
Finally, the main results are summarized and discussed in Sect. 6.

## 2. Model and data

The numerical model used in this paper is the Regional Oceanic Modeling System (ROMS) (Shchepetkin and
McWilliams, 2005). The model configuration is the same as employed in Herbert et al. (2016), and the
following text is derived from there with minor modifications.

ROMS is a three-dimensional free surface, split-explicit ocean model which solves the Navier-Stokes primitive
equations following the Boussinesq and hydrostatic approximations. We used the ROMS version developed at
the Institut de Recherche pour le Développement (IRD) featuring a two-way nesting capability based on AGRIF
(Adaptative Grid Refinement In Fortran) (Debreu et al., 2012). The two-way capability allows interactions
between a large-scale (parent) configuration at lower resolution and a regional (child) configuration at high
resolution. The ROMSTOOLS package (Penven et al., 2008) is used for the design of the configuration. The
model configuration is built following the one performed by Djakouré et al. (2014) over the Tropical Atlantic.
The large scale domain extends from 60° W to 15.3° E and from 17° S to 8° N and the nested high resolution
zoom focuses between 17° S and 6.6° N and between 10° W and 14.1° E domain. This configuration allows for
equatorial Kelvin waves induced by trade wind variations in the western part of the basin to propagate into the
Gulf of Guinea and influence the coastal upwelling (Servain et al., 1982; Picaut, 1983). The horizontal grid
resolution is 1/5° (i.e. 22 km) for the parent grid and 1/15° (i.e. 7 km) for the child grid (see Herbert et al.
(2016), their Fig. 1). This allows an accurate resolution of the mesoscale dynamics since the first baroclinic
Rossby radius of deformation ranges from 150 to 230 km in the region (Chelton et al., 1998). The vertical
coordinate is discretized into 45 sigma levels with vertical S-coordinate surface and bottom stretching
parameters set respectively to theta_s = 6 and theta_b = 0, to keep a sufficient resolution near the surface
(Haidvogel and Beckmann, 1999). The vertical S-coordinate Hc parameter, which gives approximately the
transition depth between the horizontal surface levels and the bottom terrain following levels, is set to Hc = 10





m. The GEBCO1 (Global Earth Bathymetric Chart of the Oceans) is used for the topography (www.gebco.net).
The runoff forcing is provided from Dai and Trenberth's global monthly climatological run-off data set (Dai and
Trenberth, 2002). The rivers properties of salinity and temperature are prescribed as annual mean values. One
river (Amazon) is prescribed in the parent model while five rivers, that correspond to the major rivers present
around the Gulf of Guinea, are prescribed in the child model (Congo, Niger, Ogoou, Sanaga, Volta). At the
surface, the model is forced with the surface heat and freshwater fluxes as well as 6 hourly wind stress derived
from the Climate Forecast System Reanalysis (CFSR) (horizontal resolution of ¼°x ¼°) (Saha et al., 2010). Our
model has three open boundaries (North, South, and West) forced by temperature and salinity fields from the
Simple Ocean Data Analyses (SODA) (horizontal resolution of ½°x½°) (Carton et al., 2000a, 2000b; Carton
and Giese, 2008). The simulation has been performed on IFREMER Caparmor super-computer and integrated
for 30 years from 1979 to 2008 with the outputs averaged every 2 days. A statistical equilibrium is reached after
~10 years of spin-up. Model analyses are based on the 2-days averaged model outputs from year 1998 to year
2008. The model has already been validated successfully with a large set of measurements and climatological
data, and more detailed information about the model validations can be found in Herbert et al. (2016).

For SST observations, we use data obtained from measurements made by the Tropical Rainfall Measuring
Mission microwave imager (TMI). The dataset is a merged product available at www.remss.com. The SST data
have a spatial resolution of ¼° and for the present study the 10 years' time series, from  1 January 1998 to 31
December 2008, obtained as 3-daily field. The important feature of the microwave retrievals is that it can give
accurate SST measurements under clouds (Wentz et al., 2000). However, the major limitation to the microwave
TMI observations is land contamination which results in biases of the order of 0.6°K within about 100 km from
the coast (Gentemann et al., 2010). Thus, in the Optimal Interpolation TMI product the offshore zone with no
data extends at approximately 100 km from the coast. This limits to some degree the analysis of near-coastal
regions, in particular those dominated by coastal upwelling dynamics.
We also use for this study daily sea surface height (SSH) data, which are available for the period 1993–2012 and
maintained by the organization for Archiving, Validation, and Interpretation of Satellite Oceanographic data
(AVISO; www.aviso.altimetry.fr). The sea surface height dataset is a merged product of observations from
several satellite missions Ssalto/Duacs (Segment Sol multimissions d'ALTimétrie, d'Orbitographie et de
localisation précise/Developing Use of Altimetry for Climate Studies) mapped onto a 0.25° Mercator projection
grid. All standard corrections have been made to account for atmospheric (wet troposphere, dry troposphere and
ionosphere delays) and oceanographic (electromagnetic bias, ocean, load, solid Earth and pole tides) effects.
The mean sea surface topography for the period 1993–2012 was removed from the SSH to produce sea surface
height anomalies.
In addition, surface pressure data were studied using ECMWF Atmospheric Reanalysis (ERA) for the 20th
Century product. The four-hourly data are daily averaged and is available on https://rda.ucar.edu website. The
product assimilates surface pressure and marine wind observations.






## 3. Seasonal variability of surface conditions in CLR

The purpose of this section is to describe the seasonal atmospheric and ocean surface conditions in the CLR.
The seasonal variability of SST, surface winds stress, horizontal current intensity, depth of 20° C-isotherm
(hereafter referred to as z20), and the surface net heat flux from monthly averaged model outputs in the CLR for
each year from 1998 to 2008 and averaged over the period are shown on Fig. 1. The reliability of the model is
also provided by comparing the simulated and the corresponding TMI SST climatological seasonal cycle in the
CLR (Fig. 1a). The SST variations display an annual cycle with highest temperature in boreal winter (warm
season), when the ITCZ reaches its southernmost position and the trade winds are weakest, and minimum values
in boreal summer (cold season), when the trades intensify. The most salient features of the atmospheric and
hydrographic fields during May-June are also illustrated on Fig. 1 by May-June averaged maps. Despite a warm
bias (~1° C) compared to satellite observations, well known in the eastern tropical Atlantic region (e.g. Zeng et
al., 1996; Davey et al., 2002; Deser et al., 2006; Chang et al., 2007; Richter and Xie, 2008), the model pretty
well reproduces the satellite SST pattern. The SST May-June average map indicates that the boreal summer SST
minimum is related with intensified cool SST around 6°S, in the Congo mouth region. In this region, the coast is
oriented parallel to the trade flow which reinforces in boreal summer, thus favorable to coastal upwelling
processes. The mean alongshore wind stress during May-June reveals in fact that upwelling conditions are
observed over most of the CLR. Wind stress magnitude exhibits a semi-annual variability with a second
maximum in October–December and a weakening during July-September season (Fig. 1b). The strengthening of
winds in spring is associated with a strengthening of mean current speed, particularly off Cape-Lopez between
2° S to 4° S and west of 8° E in May-June (Fig. 1c). The orientation of surface current is mostly westward for
the May-June season, while it is northward from October to January (not shown). This general picture of surface
circulation is consistent with observations (Merle, 1972; Piton, 1988; Rouault et al., 2009).
The region is also characterized by a shallow thermocline which depicts a strong semi-annual cycle (Fig. 1d).
The evolution of z20 reveals a thinning of the thermocline during May-July and a thickening up to October-
November when it exhibits a minimum.
The surface net heat flux exhibits a maximum in winter and a minimum in July (Fig. 1e), following the seasonal
cycle of solar shortwave radiations. As visible on the May-June average map, greater heating is found over cool
waters, due to weaker heat loss via latent heat flux in these areas.
The seasonal cycle is modulated by strong year-to-year variations. The mean SST in the CLR in 2005 cools as
early as March from TMI data and April from the model data. SST reaches weaker values than the climatologic
ones, as observed by Marin et al. (2009) and Caniaux et al. (2011) west of 4° E. This 2005 cold anomaly is
associated with positive wind speed and surface current speed anomaly in April-May (Fig. 1b&c) as well as
shallower-than-average thermocline depth. In 2006, SST variations are very close to the climatologic ones.


Thus, the April-June season in the CLR appears as a transitional period characterized by strong seasonal
evolution, primarily governed by the local winds which generate coastal upwelling in Congo mouth region and
modulated by the variation of thermocline depth.

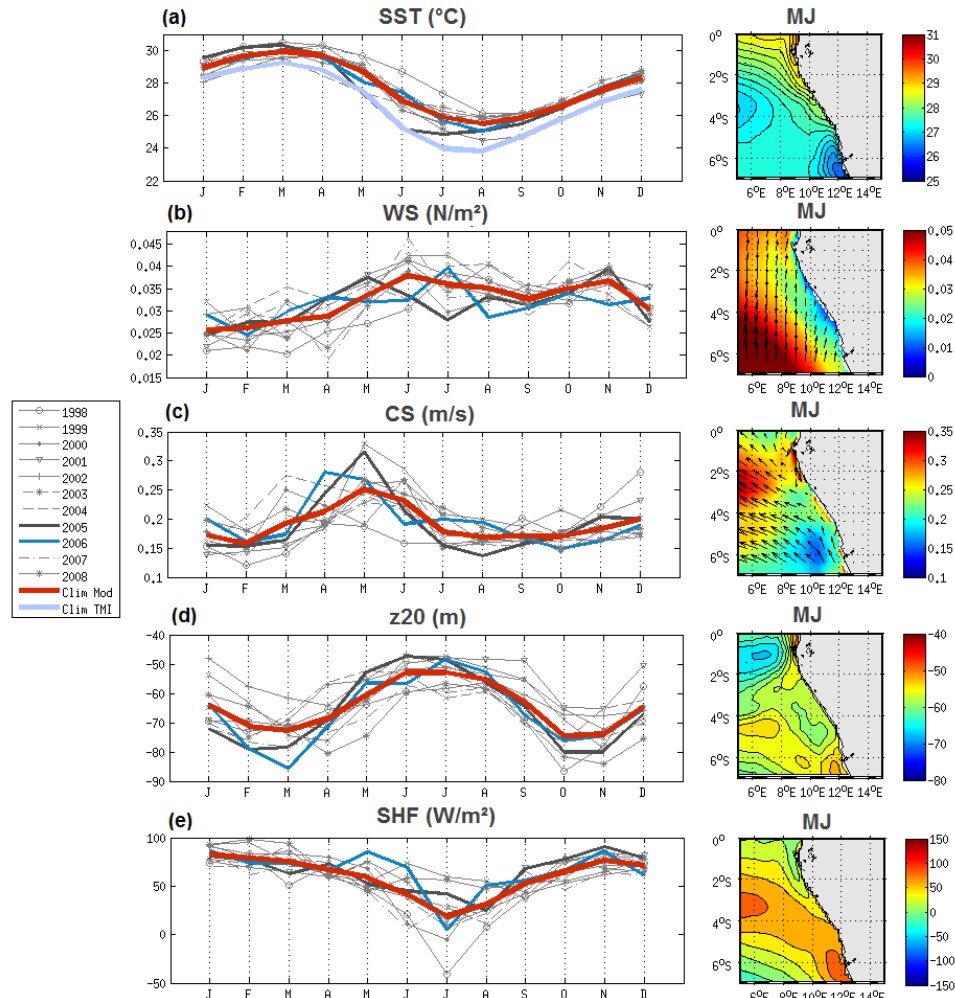


**Figure 1**: Monthly average of the (a) sea surface temperature (°C); (b) wind stress magnitude (N.m$^{-2}$); (c)
horizontal current speed (m.s$^{-1}$); (d) 20° C-isotherm depth anomalies (m); and (e) surface heat flux (W.m$^{-2}$) from
January to December from 1998 to 2008 and for the climatology (averaged over 1998-2008) simulated by the
model (red curve) and from the observations : monthly average TMI 3-daily SST data (light blue curve in (a));
averaged over 5° E-14° E and 7° S-0° S. Right panel: maps of each variable over May-June. For the wind and
the surface current, the color field shows the wind stress magnitude and the current speed respectively.




### 4. Analyze of cooling events in CLR in 2005 and 2006

In this section, we examine the impact of intraseasonal wind bursts on SST in CLR during the particular years
2005 and 2006 (Marin et al., 2009; Caniaux et al., 2011). We propose here to analyze in details the SST
conditions in CLR, east of 5° E, for both years.

### 4.1 SST variations

In order to delineate the sequence of cooling events, we analyze the SST variations from 2-days averaged model
outputs in 2005 and 2006 over the CLR, i.e. between 5° E and 12° E (Fig. 3a&4a). In 2005, the intraseasonal
cooling events took place on 22-24 April, 8-12 May, 16-20 May, 26-30 May, 12-16 June and 30 June-2 July,
with a temperature drop ranging between -0.2°C to -1.7°C. The cooling events occurred east of 5° E from May
to September. They concerned especially the southern equatorial region (around ~3-4° S), except for the
strongest events where they reached more northern equatorial regions, especially for the mid-May and late-May
2005 events. These latter were associated with an intense meridional SST front between the cold water south of
the equator and the warmer water north of the equator, as visible on SST map for 12 May 2005 presented on
Fig. 2. We can see cold waters extending along the eastern coast and in ACT region west of 5° W.  In the model,
cold waters are deflected offshore off Cape-Lopez, due to recursive bias in warm water intrusion toward the
south.
Besides, model SST fields (Fig. 3a) indicate that the SST minimum (~24° C) in 2005 was reached in July, i.e.
one month earlier than in 2006, as also noticed in seasonal variations of SST averaged in the region (Fig. 1a).
These results illustrate the important role in the CLR of the succession of quick and intense cooling events in the
establishment of persistent cold anomalies, as highlighted by Marin et al. (2009) in the equatorial region.





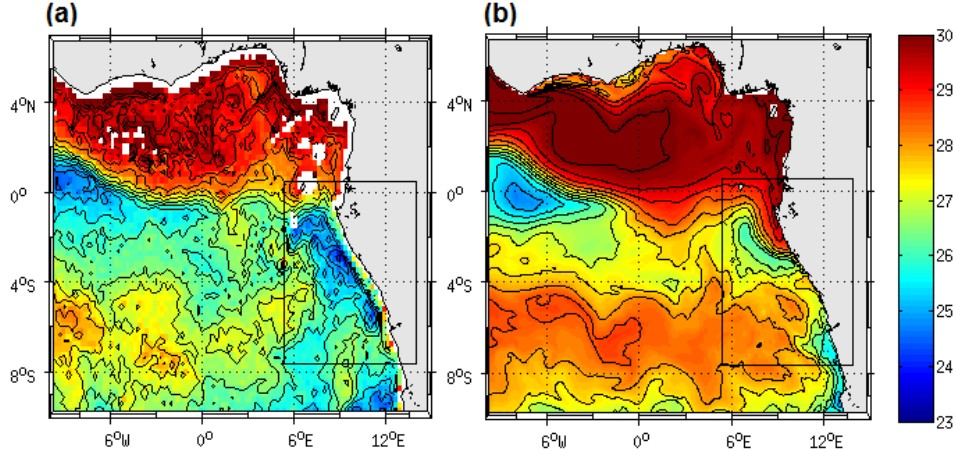

**Figure 2**: Map of the sea surface temperature (° C) on 12 May 2005 from 3-days average TMI data (a) and from the 2-days average model output (b). Note that for the model it corresponds to 11-12 May average whereas for TMI data it is 10-11-12 May average. The black square indicates the Cape-Lopez region (called 'CLR').

## 4.2 Forcing mechanisms

### 4.2.1. Local forcing

To examine the local forcing mechanisms responsible for the observed cooling events in CLR, the intraseasonal variations of wind stress magnitude anomalies are examined and compared in 2005 and 2006 (Fig. 3b & 4b).

In 2005, successive periods of 6-16 days wind intensification occurred from late-March to late-May. The main cooling events described above are associated with positive wind stress speed anomalies occurring on 12-24 April, 2-6, 12-16 & 24-28 May, 8-12 & 28 June, with a maximum for the 12-16 May event peaking on 14 May (at ~0.03 N.m$^{-2}$). Another period of wind intensification is evidenced in late March – early April but it did not generate significant cooling despite comparable or even higher wind intensity than following wind events. In 2006, periods of wind intensification were slightly shorter than in 2005 and extended from mid-March to mid-May, interrupted by periods of negative wind anomalies. The main wind events occurred in 16-18 March, 2-4 & 16-24 April, 4-6 & 12-18 May, with maximum wind stress magnitude anomaly in 16-24 April. Also, the wind event in late April 2006 did not generate a surface cooling as strong as the mid-May 2006 one, despite higher wind stress magnitude anomalies. To depict the subsurface conditions during cooling events in CLR for both years, anomalies of the 20° C-isotherm depth averaged from 5° E to 12° E are presented on Fig. 3c & 4c. They indicate strong correlation with SST anomalies on intraseasonal time scale with maximum values (up to + 25 m) observed during the 14 May 2005 event. In early April 2005 and before the late-April 2006, the thermocline was deeper, that can explain why wind intensification did not generate a surface cooling at these times. Indeed, at the





time of the strong 16-24 April 2006 wind event, the z20 anomalies was weaker south of the equator than during
the 14-16 May 2005 event, making the SST less reactive to comparable wind intensification. The same feature is
observed in early May 2006, when the z20 anomalies indicate deeper thermocline south of the equator around 4°
S than a few days later. Besides, the thermocline appeared shallower south of the equator in 2005 than in 2006,
in agreement with the difference of the cooling intensity observed between the two years.
The Ekman pumping velocity $w_e$ averaged along 3-4° S is shown on Fig. 3d. It is correlated with wind intensity,
with maximum around 8°E at the dates of the 2005 and 2006 events, in agreement with the cooling events
identified mostly around 3-4°S (Fig. 3a & 4a). The maximum upward velocity sometimes extended west of 8°
E, as during April 2005 and June 2006 events.
Another process that may contribute to the cooling in the upper layer is the vertical mixing due to intense
vertical shear of the zonal current. The maximum of the zonal current vertical shear fields in CLR, averaged
between 5° and 12° E for 2005 and 2006 (Fig. 3e & 4e), exhibited intensification south of the equator, centered
around 3-4° S. Weaker intensification also occurred occasionally at the equator (located around 80 m depth
between the westward surface South Equatorial Current – SEC – and the eastward subsurface Equatorial Under-
Current). Around 3-4° S, the vertical shear was driven by the SEC, reinforced by prevailing southerly winds
events through Ekman transport. It thus occurred at the date of wind events previously identified for 2005 and
2006, with stronger vertical shear occurring in early May 2005. The intensity of the maximum of vertical shear
during the events was quite similar between 2005 and 2006. The main difference lied in their meridional extent,
related to the meridional extent of the strengthened southerly winds which reached equatorial region during the
May 2005 events (not shown). We can also notice that for comparable wind intensification, the spring and
summer wind events were not associated with comparable intensity of vertical shear. The meridional wind
component favorable to westward Ekman transport was actually stronger during April and May events than
during summer ones (not shown).
The heat content within the mixed layer is also impacted by the sea surface heat fluxes.
The net heat fluxes averaged between 5° E and 12° E are shown on Fig. 3f & 4f for 2005 and 2006 respectively.
They indicate a net heating (~ 50-100 $W.m^{-2}$) over the 2° S - 5° S latitude band, where the SST cooling was
strongest, suggesting other mechanisms involved. However, we notice some particular events during which the
net heat flux was negative over most of the region. The strongest net cooling (-30 $W.m^{-2}$) occurred during the
26-28 May 2005 event. It was mainly due to a sudden decrease of incoming surface short wave radiation (drop
of about 140 $W.m^{-2}$ between 22 and 28 May; not shown) suggesting increased cloud cover.






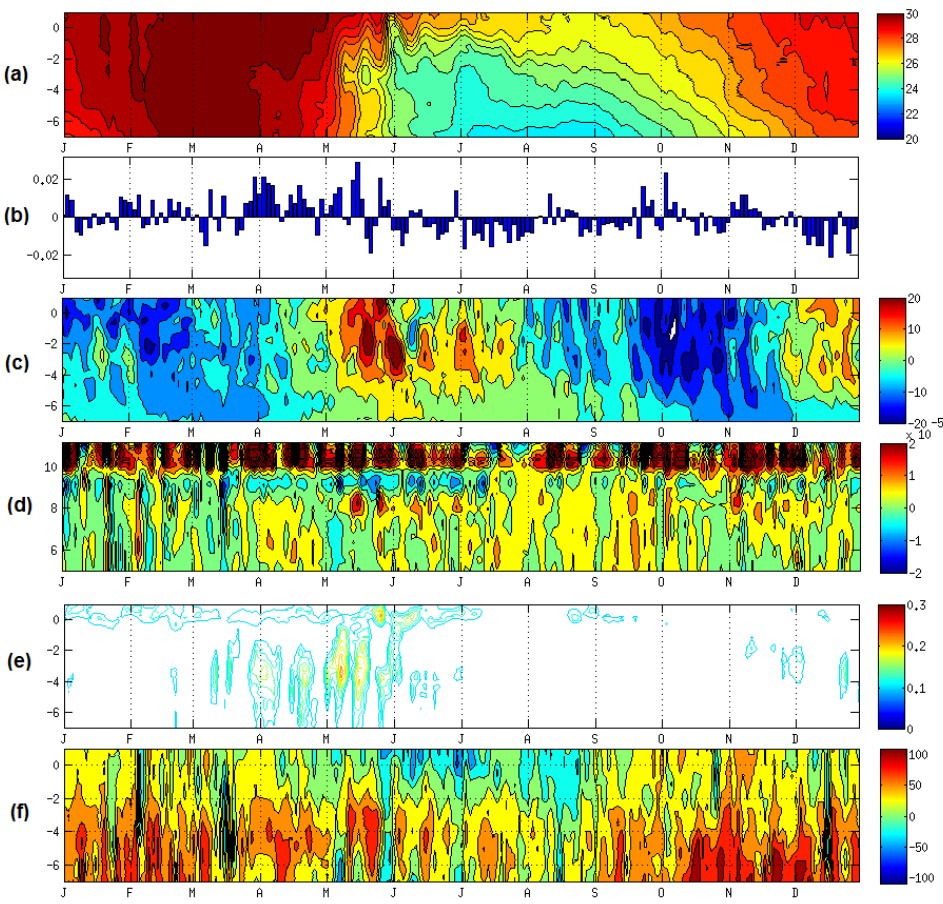

**Figure 3**: (a) Time-latitude diagram, from 7° S to 1° N, of the sea surface temperature (in ° C) averaged between 5° E and 12° E; (b) Time evolution of the wind stress amplitude anomalies (N.m$^{-2}$) averaged between 5° E and 12° E and between 3° S and 0° S; (c) Latitude-time diagram of the 20° C-isotherm depth anomalies (m) averaged between 5° E and 12° E; (d) Longitude-time diagram of Ekman Pumping anomalies (m.s$^{-1}$) averaged between 3° S and 4° S; (e) Latitude-time diagram of the maximum of the zonal current vertical shear (m.s$^{-1}$) averaged between 5° E and 12° E (only the values $> 0.1$ m.s$^{-1}$ are plotted) ; (f) Latitude-time diagram of the net heat flux (W.m$^{-2}$) averaged between 5° E and 12° E; from 1$^{st}$ Jan. to 31 Dec. 2005.






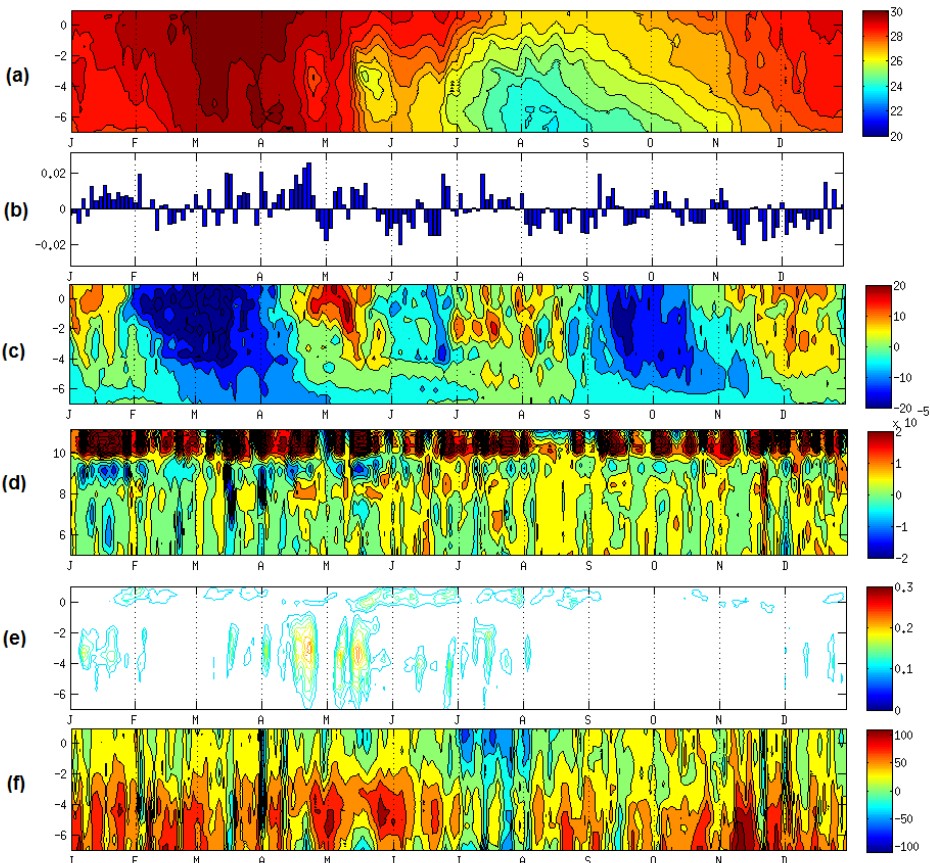


**Figure 4**: Same than Figure 3 but for 2006.


### 4.2.2. Remote forcing

### a. Highlighting of Kelvin wave propagation

As previously shown, the time of occurrence of the cold events in the CLR coincides with steeper thermocline
slope which allows a mixed layer temperature to be more reactive to surface forcing. Indeed, because of its
proximity to the equator, the thermocline in CLR is affected by the arrival of equatorial waves, initiated in the
west part of the basin. Pairs of alternate downwelling and upwelling Kelvin waves occur usually in February-
March, July-September and October-November. Upon impingement with the eastern boundary, the incoming
equatorial Kelvin wave excites westward-propagating Rossby waves and poleward-propagating coastal Kelvin
waves (Moore, 1968; Moore and Philander, 1977; Illig et al., 2004; Schouten et al., 2005; Polo et al., 2008). The



20° C-isotherm depth anomalies along the equator and along 9°E are presented on Fig. 5 and clearly evidence
large positive anomalies indicating shallower-than-average thermocline, propagating eastward along the equator
and then southeastward for both years. The eastward propagation of Kelvin wave along the equator and
southeastward along the coast is also well visible in the basin-wide SSH anomalies (Fig. 6) with a phase
velocity of about 1.1-1.3m.s$^{-1}$, which fits well in the range between the second and third baroclinic equatorial
Kelvin wave modes. The upwelling wave is in fact detectable by negative SSH anomalies, associated with
shallower-than-average thermocline. In 2005, negative (positive) SSH (z20) anomalies occurred in the West in
early April and in mid-May, whereas they occurred around late-March and early May in 2006. The first Kelvin
wave thus reached the CLR slightly earlier in 2006 than 2005, at the beginning of May.
Thus, the intensity of the cold events observed in spring and summer 2005 and 2006 resulted from both the
basin preconditioning by remotely forced shoaling of the thermocline, local mixing and upwelling processes in
response to strong southerly local winds, as well as heat flux variations. In 2005, stronger wind intensification
and favorably preconditioned oceanic subsurface conditions, made the coupling between surface and subsurface
ocean processes more efficient than in 2006, resulting in stronger cooling.

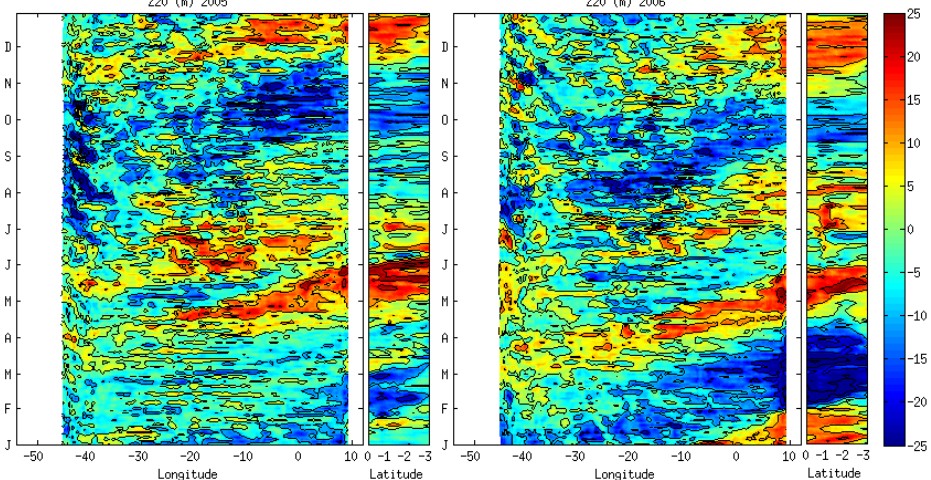


**Figure 5**: Time evolution of the 20° C-isotherm depth anomalies (m) along the equator (between 54° W and 12°
E) and along 9° E (between the equator and 3° S) for 2005 (left) and 2006 (right).



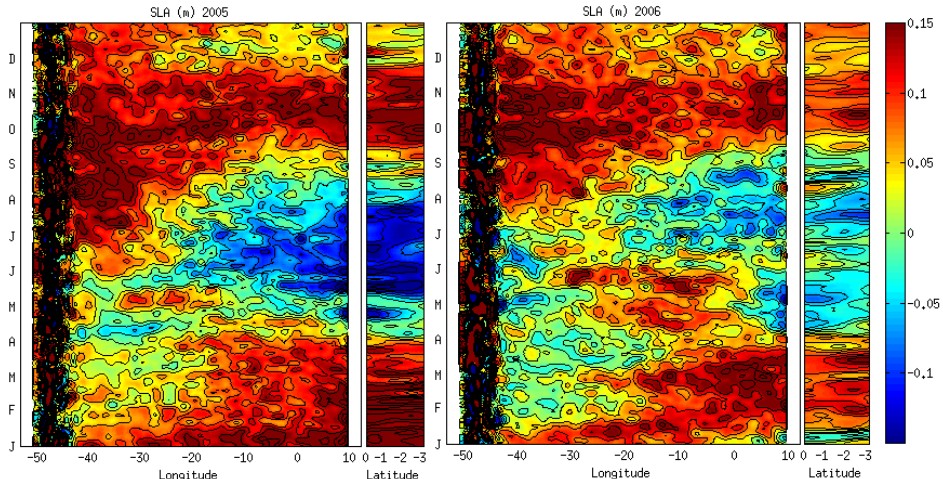

**Figure 6:** Time evolution of the sea level anomaly (m) along the equator (between 54° W and 12° E) and along
9° E (between the equator and 3° S) for 2005 (left), and 2006 (right) from AVISO data.

**b. Kelvin wave generation and coinciding atmospheric conditions in the West**

In order to identify the wind activity which accompanies the excitation of Kelvin upwelling waves in winter
2005 and 2006 in the west part of the basin, we analyze the position of the ITCZ (averaged over 50° W-35° W)
identified as the latitude where the meridional wind stress goes to zero (Fig. 7a). The anomaly of zonal and
meridional components of the wind stress (Fig. 7b&c), the wind stress curl anomaly (Fig. 7d), as well as the z20
and SSH anomalies (Fig. 7e&f), averaged in the equatorial band (over 1° S and 1° N), are also presented. For
the zonal wind stress, the frequencies > 1 month are removed by a low-pass filtering and superimposed with the
anomalies (red curve on Fig. 7c). Many authors suggest that the source of the equatorial Kelvin wave is mainly
related to a sudden change of the western equatorial zonal wind (e.g. Picaut, 1983; Philander, 1990): a
symmetric westerly (easterly) wind burst along the equator will generate Ekman convergence (divergence) and
thus force downwelling (upwelling) anomalies which then propagate eastward as a Kelvin wave (Battisti, 1988;
Giese and Harrison, 1990). In 2005, shallower-than-average thermocline, evidenced by positive (negative) z20
(SSH) anomalies, occurred in the end of March-beginning of April in the west part of the basin (Fig. 7e&f). The
meridional and zonal wind stress anomalies indicate that the maximum of thermocline slope anomaly was
associated with a strengthening of northeast trades followed by a strengthening of southeast trades from either
side on the equator. At the equator, we notice indeed a sudden reversing of meridional winds which turned
southward on 27-28 March 2005 related to an abrupt southward displacement of the ITCZ which was then found
south of the equator in the west part of the basin (Fig. 7a&b). The ITCZ returned its initial position four days
later followed by a strengthening of easterlies which persisted for ~20 days (Fig. 7c). Climatologically, the




latitudinal position of the ITCZ varies from a minimum close to the equator in boreal spring (March-May) in the
west to a maximum extension of $10°N – 15°N$ in late boreal summer (August) in the east. Positive (negative)
wind stress curl is found north (south) of the ITCZ. When the ITCZ is north of the equator, it induces upward
(downward) Ekman pumping to the north (south) of the ITCZ. Thus, the southward shift of the ITCZ on 27-28
March 2005 accompanied with strong northerlies led to negative anomaly of wind stress curl south of the
equator resulting in upward Ekman pumping. Results show indeed a strong negative anomaly on 22-26 March
2005 associated with the southward shift of the ITCZ just before the upwelling signal, initiated on 28 March.
These changes contributed to a rise in the oceanic thermocline with a time lag of some days (Fig. 7e&f). The
upwelling signal might then be reinforced by the symmetric easterly wind which concerned a large part of the
western basin. Besides, we identify on Fig. 6d another peak of negative wind stress curl anomaly on 6-8 May
2005, more sudden than the previous winter one. It was associated with positive (negative) z20 (SSH) anomaly
indicator of a thermocline rise initiated on 6 May 2005 in the west of the basin and which propagated eastward
along the equator. The zonal wind stress anomalies (Fig. 6b) also indicate an easterly wind strengthening
initiated in the beginning of May, which a maximum on 8-10 May, just after the minimum of wind stress curl.
In 2006, the upwelling Kelvin wave is identified in the first half of March in the west part of the basin (Fig.
7e&f). The coinciding atmospheric conditions were slightly different than the ones identified in 2005. In winter,
the position of the ITCZ had a more southern position in 2006 than in 2005. It crossed the equator during a
longer period (about 10 days from ~ Feb. 10 2006), reaching minimum latitude on 22-24 February. This location
south of the equator induced a negative wind stress curl anomaly (Fig. 7d). As in 2005, the reversion of the
meridional wind at the equator was followed by a strengthening of westward component of the wind stress few
days after, which lasted for about ten days (Fig. 6c); however, it was of a lesser magnitude compared to 2005
and only concerned the westernmost part of the basin. In addition, the negative zonal wind anomaly concerned
mainly the northeasterlies rather than the southeasterlies, leading to an anti-symmetric meridional wind pattern as
well as symmetric zonal wind pattern on either side on the equator (not shown). These wind patterns were
expected to generate Ekman convergence at the Equator and thus to reinforce the observed upwelling anomalies.

Thus, for both years, Kelvin upwelling wave occurred in the west while easterly winds were strengthened from
either side of the equator after the ITCZ reached its southernmost location. This latter was observed one month
earlier in 2006 than in 2005, and was associated with a negative wind stress curl anomaly. In winter 2005, the
ITCZ was found south of the equator after a very sudden southward shift and was followed by strong easterlies
during ~20 days, while in winter 2006, the ITCZ was found closer to the equator less sharply and during a
longer period, followed by weaker easterlies compared to 2005.





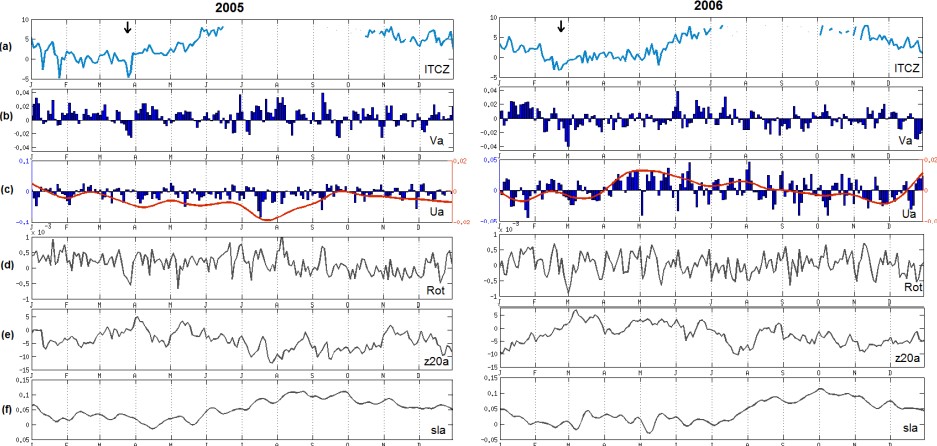


**Figure 7**: Time evolution, from 2-days averaged model outputs, of (a) the position (in latitude, between 5° S and 10° N) where the meridional wind stress value equal zero (indicator of the position of the ITCZ), over Jan-Dec 2005 (left) and Jan-Dec 2006 (right); (b) the meridional wind stress anomalies (N.m$^{-2}$) averaged between 50° W and 35° W and between 1° S and 1° N; (c) same as (b) but for zonal wind stress anomalies (N.m$^{-2}$) (in blue). The red curve is after the frequencies > 1 month are removed by a low-pass filtering; (d) the wind stress curl anomalies (N.m$^{-2}$) ; (e) the 20° C isotherm depth anomalies (m); (f) the sea level anomalies (m). The black arrow in (a) indicates the southward shift of the ITCZ before the excitation of the Kevin wave (see text).


**4.3. Westward extension of the CLR cooling**
In the east, the cooling generated by southerly wind bursts in the CLR then progressively extended westward to
connect with the southern boundary of the equatorial ACT. This phenomenon was more obvious in 2005 when
the cooling which first concerned coastal area extended further offshore a few days after the two strong events
occurring in the second half of May. To evidence the effect of these events on SST, maps of SST anomalies and
wind stress monthly anomalies averaged over May 2005 and over the weeks before the strong events (from 1 to
10 May) are presented on Fig. 8. The results illustrate an enhancement after 10 May of the cooling in the east
associated with southerly wind intensification and an extension of the cooling especially south of the equator up
to 20°W.








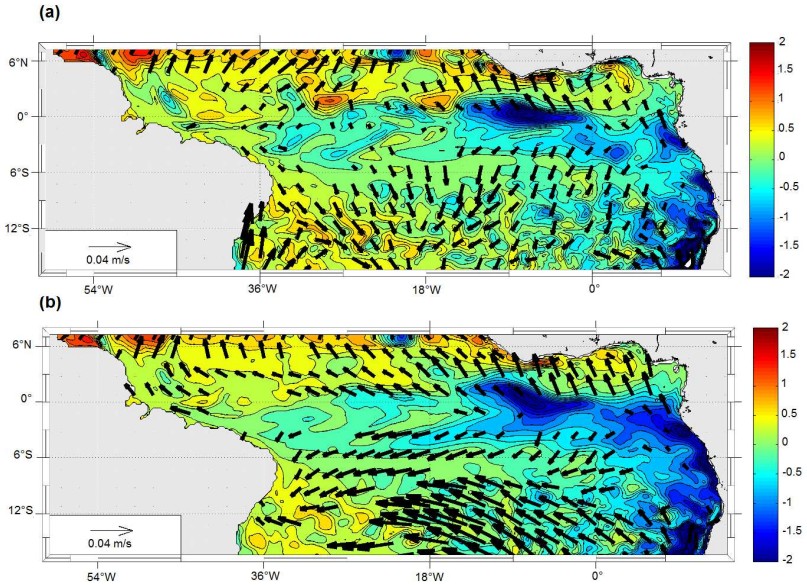

**Figure 8**: (a) Sea surface temperature anomalies (° C; color) superimposed with wind stress intensity anomalies (arrows) averaged over May 2005. Only the values > 0.02m.s$^{-1}$ are plotted; (b) same but averaged between 1 May and 10 May 2005 (b)

To better understand the oceanic processes implied in this cooling extension, we examined the z20 anomalies, SSH anomalies and zonal velocities along 3° S (Fig. 9 b-d). They reveal that the cooling westward extension was associated with a westward propagation of a steeper thermocline and negative SSH anomalies from the African coast up to 5°-10° W combined with enhanced surface westward current fluctuations at the dates of the successive events from April-June. The fluctuations of the westward surface current occurring off Gabon with periods of ~8-10 days were related to the strengthening of southerly winds during the wind bursts at the same periods (Fig. 3b & 4b). The surface current in this area is part of the westward SEC which is known to intensify during the cold season (Okumura and Xie, 2006). Our study implies shorter time scales than seasonal scale but the intensification of the SEC during wind bursts through Ekman transport processes might contribute to the westward extension of the cooling by advection of cold eastern upwelled water. This is in agreement with DeCoëtlogon et al. (2010) who found from model results that at short time scale (a few days), more than half of the cold SST anomaly around the equatorial cooling could be explained by horizontal oceanic advection controlled by the wind with a lag of a few days. In addition, the z20 and SSH show respectively negative and positive anomalies propagating westward at 3° S (Fig. 9), initiated from the coast with a propagating speed of around 10 cm.s$^{-1}$, which is very close to the phase speed of Rossby waves. Indeed, the excitation of the westward waves at the coast coincided with the arrival of Kelvin waves (see Fig. 5) suggesting the possibility of Kelvin wave's reflection processes into symmetrical westward propagating Rossby waves. A westward



propagation of z20 and SSH anomalies, although less obvious, was presently also identified at 3° N (not shown).
In 2006, no similar wave generation process is observed (not shown). In 2005, the locally wind-forced
component of the wave might reinforce the remote part of the reflected wave signal at the coast by the sea level
slope which balanced the strengthening of alongshore winds blowing during the mid-May and late-May events.
The quantitative and respective contributions of local and remote wind forcing to this wave is out of the aim of
this study and would require further analysis.
Thus, the combined effects of westward surface currents (via advection and vertical mixing through horizontal
current vertical shear), local wind influences (via vertical mixing) and wave westward propagation, resulted in
the extension of cold upwelled water from the eastern coast to near 20° W.

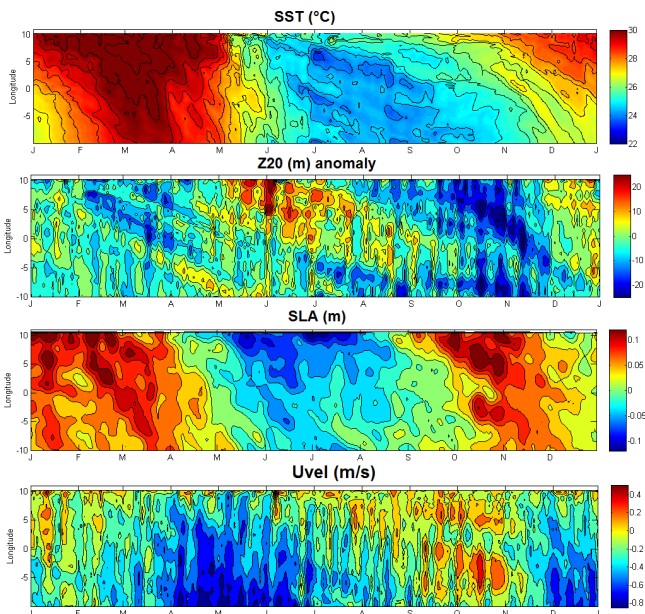


**Figure 9**: Time-longitude diagrams at 3° S between 10° W and 10° E, and from 2-days averaged model outputs

from January to December 2005, of (from top to bottom) i) the sea surface temperature (° C); ii) the 20° C

isotherm-depth anomalies (m); iii) the sea level anomalies from AVISO data (m); and iv) the zonal component

of surface velocity (m.s$^{-1}$).


**5. Focus on the mid-May 2005 event**
We have previously identified five main cold events in 2005 (22-24 April, 8-12 May, 16-20 May, 26-30 May
and 14-18 June), characterized by a temperature drop ranging from -0.2° C to -1.7° C in the model. Analysis of





wind stress magnitude has revealed that each event is associated with strengthening of equatorward winds,
especially during the 14-16 May event when the wind stress magnitude anomaly averaged over the CLR is the
strongest one. This particular event has been found to be responsible for the sudden and intense SST cooling in
the eastern equatorial Atlantic and identified as part of manifestation of temporal variability of the St. Helena
Anticyclone (Marin et al., 2009). In this section, we focus on this mid-May event, to better understand the
processes in play during this unusual event.

**5.1 Atmospheric conditions**

**5.1.1 Wind and surface atmospheric pressure**
The spatial distribution of the mid-May 2005 wind event can be inferred from Fig. 10 where anomalous CFSR
wind speed fields superimposed with daily precipitation fields, surface pressure, wind speed curl, and downward
shortwave radiation, are presented from 13 May to 17 May. The event was characterized by intense
southeasterly wind anomalies east of 15° W and from 30°S to the equator from 13-14 May, concomitant with a
strengthening of the easterlies west of 30° W between 30° and 15° S (Fig. 10a). The wind anomaly extended
then westward up to 15-16 May when the maximum was located in the western part of the basin off northeastern
Brazilian coast. Simultaneously, a strengthening of southerly winds occurred north of the equator in the Gulf of
Guinea. The anomalous strong winds during the event were associated with anomalous high pressure core of the
Saint Helena Anticyclone, especially on 13-14 May, also associated with particularly low pressure under the
ITCZ.



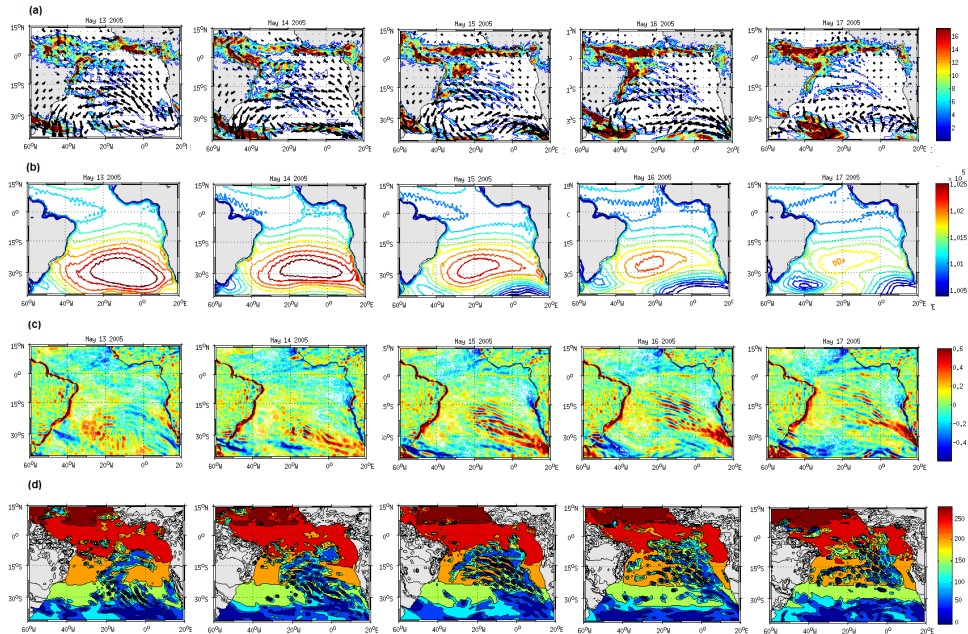

**Figure 10**: Daily-averaged, from 13 May to 17 May 2005 (left to right panels), of (a) precipitation rate (kg.m$^{-2}$/day) (color field) superimposed with wind speed anomalies (vectors) (m.s$^{-1}$) from CFSR fields; (b) surface pressure (hPa) from ERA-20C reanalysis; (c) wind speed curl (m.s$^{-1}$) computed from CFSR wind speed fields; and (d) downward short-wave radiation (W.m$^{-2}$) from CFSR fields.

### 5.1.2 Precipitation

The maps of precipitation rate during the event (Fig. 10a) display a band of heavy precipitation (9-17 kg.m$^{-2}$/day) between 5° - 9° N and off northeast Brazil from the coast to 15° W and from 10° S to 3° S. The maximum precipitation rate in this region occurred on 15-16 May concomitant with the easterly winds strengthening. This convective zone, located between the ITCZ north of the equator and the South Atlantic Convergence Zone (SACZ) in southern tropics, is the Southern Intertropical Convergence Zone (SICZ) (Grodsky and Carton, 2003). This zone forms usually later, by June-August, when the southern branch of the convection separates from the ITCZ which moves north of the equator. Grodsky and Carton (2003) showed that this rainfall pattern appears closely linked to the seasonal change in SST difference between the ACT region (which they defined between 15° W – 5° W, 2° S – 2° N) and the SITCZ region (25° W - 20° W, 10° S - 3° S). They argued that the seasonal appearance of the ACT along the equator sets up pressure gradients within the boundary layer that induce wind convergence in the SITCZ region. Based on Grodsky and Carton (2003) results, the unusually



rainfall conditions during mid-May event might thus be explained by strong SST gradient between the two
regions caused by unusually early cooling in the ACT region at this time of the year.

**5.1.3 Generation of atmospheric gravity wave**

The precipitation fields during the mid-May event (Fig. 10a) also evidence rainfall pattern typical of
atmospheric gravity wave train characterized by a horizontal wave length ~500 km and initiated by a front
system (forming the northern boundary of a low pressure system) which developed around 17° S on 14 May and
traveled northeastward until 17 May. The rainfall train was associated with oscillatory wind stress curl train
alternating between positive and negative anomalies (Fig. 10c) as well as alternating downward shortwave
radiation minimum (Fig. 10d) associated with the wave clouds. Gravity waves are known to play an important
role in transporting the momentum and energy through long distances (Fritts, 1984). Here, they would be a way
to carry momentum and energy from South Atlantic to the equator during the strong event. To estimate the
effect of the passage of the wave on SST, the SST TMI fields have been analyzed. Given the time resolution of
the dataset (3-days averaged) a propagating train in SST fields cannot be highlighted; instead, 3-daily TMI SST
averaged (available east of 10° W) differences, as well as 3-daily averaged downward shortwave radiation
(hereafter DSWR) fields differences, between 13-14-15 May averaged and 10-11-12 May averaged and between
16-17-18 May averaged and 13-14-15 May averaged. Results indicate a good correlation between the two fields,
suggesting a northeastward propagation of the cooling associated to northeastward propagation of the DSWR
minimum associated with the displacement of the atmospheric gravity wave. A cooling first occurred south of
24° S associated with DSWR minimum, and then reached more northern region concomitant with the northward
migration of DSWR. Thus, these results suggest that the arrival of atmospheric gravity wave initiated in South
Atlantic following atmospheric disturbance would contribute to the cooling observed in the Gulf of Guinea at
the time of the mid-May event.





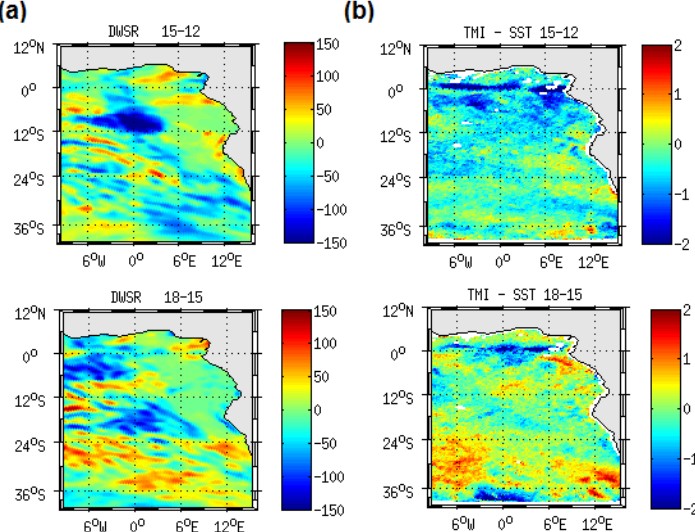


**Figure 11**: Three daily-averaged differences, between 10° W and 15° E and between 40° S and 12° N from 15-
12 May and 18-15 May 2005 (the dates indicated in the titles correspond to the last day taken into account for
the calculation of the average. e.g.: 'SST 15-12' corresponds to the difference between the mean of SST over
13-14-15 May and over 10-11-12 May) of downward short-wave radiation (W.m⁻²) from CFSR fields (a); and
SST from TMI data (b);

**5.2 A decisive event for coastal monsoon onset**

In order to analyze the air-sea pattern in the northern Gulf of Guinea during May-June 2005, we show on Fig. 12
the wind stress magnitude, precipitation rate, and SST fields averaged from 10° W to 6° W. The wind
strengthening appeared first south of the equator on 12-16 May and then north of the equator from 14-18 May. It
was associated with strong rainfall extending southward up to 2° N. Equatorial cooling occurred 4 days after
the event and slowed down the overlying winds by feedback mechanisms. The winds north of the equator then
remained stronger than in the ACT region and strengthened again north of the Equator on 22-28 May together
with precipitation maximum pushed northward (around 5° N) after the event.
Thus, this mid-May event appears as the "decisive event" which triggered the abrupt transition between the two
wind patterns in the northern Gulf of Guinea, when the wind north of the equator became and remained stronger
than south of the equator. It occurred 15 days earlier than the average date (31 May) identified by Leduc-
Leballeur et al. (2013) over 2000-2009 period. According to these authors, the time of occurrence of this
phenomenon would be related with the strength of anomalous moisture flux. They explain that in April-May the
low atmospheric local circulation is present only during an equatorial SST cooling and surface wind





strengthening north of the equator, both generated by a southerly wind burst, before disappearing until the next
wind burst. In June-July the low atmospheric local circulation is then always present and intensified by the wind
bursts. Thus, the establishment of an abrupt seasonal transition event as observed in 2005, occurring much
earlier than the reference date, supposed anomalously strong equatorial cooling caused by unusual strong
southerly winds which allowed, through air-sea interactions mechanisms, to trigger the deep atmospheric
convection in the Gulf of Guinea at a self sustaining level.

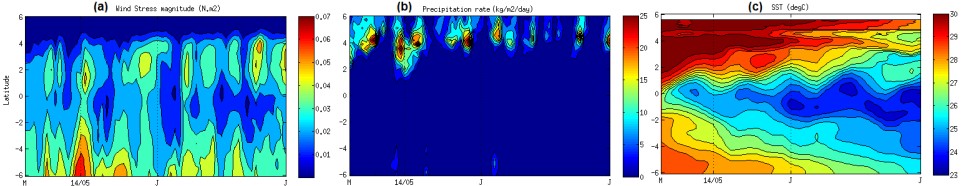

**Figure 12**: Time evolution, in May and June 2005 between 6° S and 6° N and averaged between 10° W and 6°
W, of the (a) daily averaged wind stress magnitude (N.m$^{-2}$) computed from CFSR wind speed fields ; (b) daily
averaged precipitation rate (kg.m$^{-2}$/day) from CFSR fields and (c) 2-daily averaged SST (° C) fields, from the
forced model.

**5.3. Why the mid-May 2005 event was so singular?**

To better understand which makes the particularity of the mid-May 2005 event, the atmospheric and oceanic
conditions (wind stress magnitude, short-wave radiation flux (hereafter RADSW), z20, SST, and meridional
SST gradient) averaged over the 10° W-6° W region and between 15° S to 5° N during April-May are analyzed
along the 1998-2008 period (Fig. 13). The wind stress magnitude during mid-May event appears to be one of the
strongest over the whole 1998-2008 period. These strong wind conditions are usually met later in late boreal
spring or summer, when the St. Helena Anticyclone strengthens and shifts northward toward the warm
hemisphere. The wind intensification in mid-May 2005 was associated with particularly weak RADSW from
South Atlantic to the northern equatorial region, suggesting cloud albedo effect during the event which tended to
cool the mixed layer. We can notice that the April-May 2005 period was characterized by the weakest mean
RADSW.
In addition, at the time of the event, the surface waters were already cooled by previous wind bursts (e.g. 20
April and 8 May). The SST response to the mid-May event occurred 4-6 days later, inducing the weakest
equatorial SST values for April-May season over the whole 1998-2008 period. The cooling also caused an
enhanced SST front around 1° N, as shown on Fig. 13 (bottom panel), which was found to be the earliest and
strongest one over the 1998-2008 period. This meridional SST gradient was responsible for the wind surface
intensification north of the equator (Fig. 12 and 13a) through air-sea interaction mechanisms as described by
Leduc-Leballeur et al. (2011). Another SST gradient maximum is found at the end of May 1998 but it was not
extended as eastward than during the mid-May 2005 event (not shown).



When the wind burst occurred on 14 May 2005, the 20°C-isotherm depth in the area was shallow south of the
equator and slightly deeper at the equator (Fig. 13c). The thermocline shoaling associated with the Kelvin wave
appeared in fact a few days earlier providing favorable subsurface conditions which made the SST response to
previous wind bursts (20 April and 8 May) more effective. At the time of the mid-May event, the wave already
reached more eastern areas, as shown in previous sections.

The mid-May 2005 event was also characterized by a particularly low surface pressure under the ITCZ, as
shown on Fig. 14a which displays the surface pressure north of the equator averaged between 45° W and 20° W
for April-May over the 1998-2008 period. The pressure fall during the mid-May 2005 event appeared as the
lowest in May over the whole decade. It coincided with particularly high surface pressures in St. Helena
Anticyclone region 4 days earlier (Fig. 14b). The meridional surface pressure gradient during the event is thus
found to be the strongest over 1998-2008 period (Fig. 14c). That suggests strong Hadley circulation intensity
during the mid-May event and therefore strong anomalous equatorward moisture flux, allowing the deep
atmospheric convection in the Gulf of Guinea to be triggered at a self-sustaining level, as previously described
in Sect. 5.2.

Thus, the particularity of the mid-May 2005 event mainly lies in the i) anomalous atmospheric conditions
related to anomalous strong St. Helena Anticyclone perturbation; ii) cooling initiated by the succession of
previous wind bursts; and iii) favorable subsurface local ocean conditions preconditioned by equatorial waves
which shoaled the mixed layer. Another wind burst of comparable intensity occurred at the beginning of May
2000 while the thermocline was shallow, causing SST cooling at the equator (Fig. 13&14). However, the wind
strengthening was less sudden than during the mid-May 2005 event and the resulting cooling took place over a
less broad region (not shown). In addition, the surface pressure drop in the ITCZ region was not as pronounced
as during mid-May 2005 event.





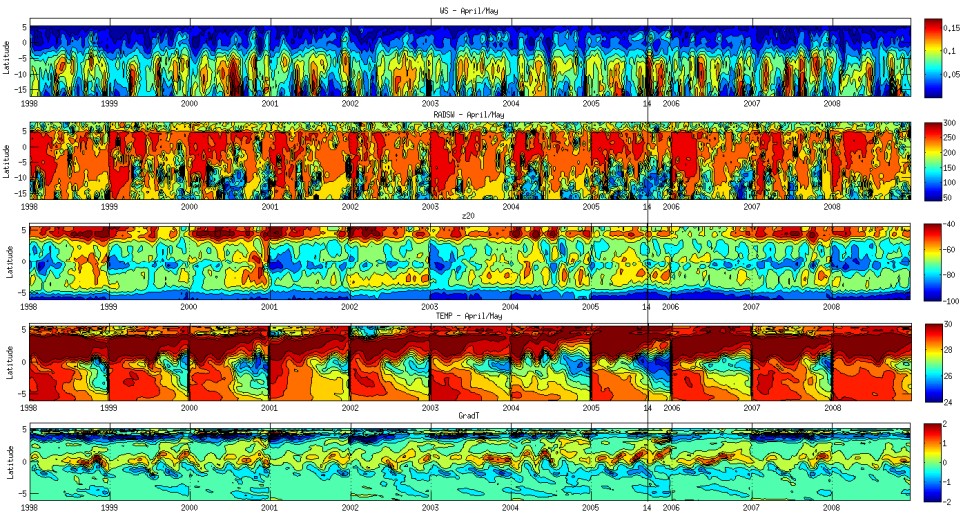


**Figure 13:** Time-latitude diagrams for April-May along the 1998-2008 period, of 2-days average, from top to
bottom, i) wind stress magnitude (N.m$^{-2}$) from CFSR fields; ii) short-wave radiation surface flux (W.m$^{-2}$) from
CFSR fields; iii) 20°C isotherm depth (m) computed from the forced model SST; iv) SST (°C) and v)
meridional SST gradient (every 0.5° of latitude), from the forced model; averaged over 10° W-6° W. The
vertical black line indicates the date of 14 May, 2005.

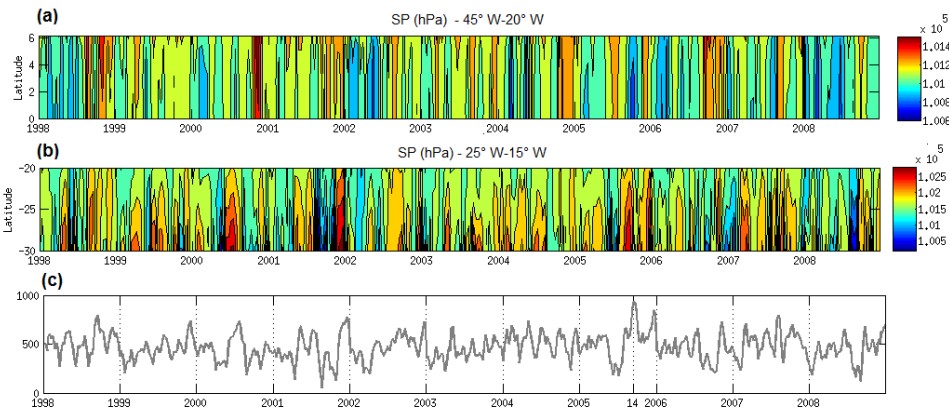


**Figure 14:** Time-latitude diagram, for April-May over the 1998-2008 period, of the surface pressure (hPa) from
ERA-20C reanalysis; (a) from 0° N to 6° N averaged between 45° W and 20° W ; and (b) from 30° S and 20° S
averaged between 25° W and 15° W; (c) differences between (a) and (b);.





### 6. Summary and discussion

In this study, the impact of intraseasonal winds on SST in the far eastern Tropical Atlantic during boreal spring 2005 and 2006 has been investigated observations and numerical simulation. We first focus our study in Cape-Lopez region (CLR), east of 5°E and between the equator and 7° S, where the seasonal and interannual SST variability is poorly documented. There, the boreal spring (AMJ) season corresponds to a transitional period between high SST in boreal winter and weak SST in boreal summer, under the influence of local winds. Intensified cool SSTs are observed in the coastal upwelling area located around 6° S in the Congo mouth region, associated with mean alongshore wind conditions. Spring season is in fact characterized by maximum winds amplitude, influence of which is made more effective by shallow thermocline depth, itself strongly influenced by remote forcing. The seasonal cycle in the CLR is modulated by strong year-to-year variations, as observed in spring 2005 when cold SST anomaly are associated with shallower-than-average thermocline depth and positive wind speed anomaly.

The intraseasonal wind bursts which occurred in spring 2005 and 2006 generated cooling events especially around 3°-4° S but for some strongest events when the cooling reached more northern equatorial region, especially during the mid-May and end-May 2005 events. The intensity of the cold events resulted from both basin preconditioning by remotely forced shoaling of the thermocline (via Kelvin wave), local mixing (induced by current vertical shear) and upwelling processes in response to strong southerly local winds. For some particular event, as the 26-28 May 2005, the net heat flux also tended to cool the surface water, due to enhanced cloud cover which decreased the incoming solar radiations. In CLR, stronger wind intensification and favorably preconditioned oceanic subsurface conditions in 2005 made the coupling between surface and subsurface ocean processes more efficient than in 2006, resulting in stronger cooling.

The preconditioning of subsurface conditions in the area via Kelvin wave at the dates of the wind bursts depended on the atmospheric conditions in the western part of the basin a few weeks earlier. Previous studies (e.g. Picaut, 1983; Philander, 1990) suggest that the source of the equatorial Kelvin wave is mainly related to a sudden change of the zonal wind in the west. Analysis of atmospheric and oceanic conditions at intraseasonal to daily scale in winter 2005 and 2006 showed that for both years, Kelvin upwelling wave was initiated in the west while easterly winds were strengthened from either side of the equator just after the ITCZ to be at its southernmost location. This latter was observed one month earlier in 2006 (late February – early March) than in 2005 (late March-early April), and was associated with a negative wind stress curl anomaly. In winter 2005, the ITCZ was found south of the equator after a very sudden southward shift and was followed by strong easterlies during ~20 days, while in winter 2006, the ITCZ was found closer to the equator less sharply and during a longer period, followed by weaker easterlies when compared to 2005. These results obtained for 2005 and 2006 years do not imply that same atmospheric conditions would be observed for winter upwelling Kelvin wave of other years. Especially, the year 2005 was very particular and also exhibited anomalously cold SSTs in the south Atlantic and anomalously warm SSTs in the north Atlantic initiated in fall 2004, signature of a meridional mode (Virmani and Weisberg, 2006; Foltz and Mc. Phaden, 2006; Hormann and Brandt, 2009).





Upon impingement with the eastern boundary, the incoming equatorial Kelvin wave excites westward-propagating Rossby waves and poleward propagating coastal Kelvin waves. In 2005, the Kelvin wave reached the coast around mid-May while southerly winds strengthened, allowing the reflected wave to be reinforced by the local wind. This resulted in westward propagation of positive (negative) z20 (SSH) anomalies which, combined with enhanced westward surface currents, provided favorable conditions to westward extension of cold upwelled water from the eastern coast to near 20°W through advection and vertical mixing.

In the second part of the study, we specially focused on the mid-May 2005 event (13 May to 16 May) that was characterized by strong southerly wind strengthening in the eastern Tropical Atlantic Ocean. It was found to be responsible for the sudden and intense SST cooling in the Gulf of Guinea and the CLR, and involved in the early onset of the ACT development in 2005 and therefore in early onset of the WAM. The analysis of atmospheric and oceanic conditions in the Gulf of Guinea associated to this event allowed to show that the mid-May event, controlled by the St. Helena Anticyclone, can be identified as a "decisive event" which triggered the abrupt transition between two wind patterns in the northern Gulf of Guinea. Unusual strong southerly winds induced anomalously strong equatorial cooling which in turns slowed down the overlying wind feedback mechanism and generated stronger than normal southerlies north of the equator through the SST front around 1°N. This triggered the deep atmospheric convection in the Gulf of Guinea at a self-sustaining level and the beginning of coastal precipitation. The time of occurrence of this phenomenon, 15 days earlier than the averaged date (31 May from Leduc-Leballeur et al., 2013), suggests than the mid-May 2005 event was associated with anomalous strong moisture flux. The description of atmospheric conditions over the 1998-2008 period has shown that the 2005 event was characterized by the strongest surface pressure gradient between the St. Helena high pressures and the low pressures under the ITCZ, inducing anomalous strong Hadley cell activity. No similar atmospheric pattern was observed during the whole 1998-2008 period. Another wind burst of comparable wind intensity occurred at the beginning of May 2000. This event also induced a cooling at the equator but the surface pressure decrease in ITCZ region was not as pronounced than during mid-May 2005 event and the SST gradient around 1° N was weaker. In addition to coastal precipitation in the Gulf of Guinea and due to the early cooling in the ACT region, unusually rainfall conditions also occurred between the northeast coast of Brazil and 15° W within the SITCZ, which generally forms in early boreal summer.

Finally, this study highlights the impact of a strong southerly wind burst in the eastern tropical Atlantic during boreal spring season, which is a transitional period during which an anomalous strong energy input may tip the energy balance from an equilibrium state toward another one and thus impact the WAM system. The analysis of atmospheric and oceanic conditions during the mid-May 2005 wind event allows to highlight the different processes through which the wind power provided by the wind burst is brought to the ocean: i) direct effect of the wind on the SST in the eastern tropical Atlantic, ii) energy transport via atmospheric gravity waves from South Atlantic, and iii) energy supply to Rossby wave. In addition to unusual atmospheric conditions in mid-May 2005, the ocean response intensity to this event was also enhanced by the subsurface conditions, made favorable by previous wind bursts, either local (e.g. in 6-8 May) or occurring a few weeks before in the West.



It is crucial to better describe the atmospheric and oceanic processes in play during such extreme event, notably
in order to reduce the well known warm bias in the southeastern tropics in coupled models in both atmospheric
and oceanic components (Zeng et al., 1996; Davey et al., 2002; Deser et al., 2006; Chang et al., 2007; Richter
and Xie, 2008). This warm bias is well evidenced in our numerical simulation (Fig. 1&2) and our results clearly
show that the cooling events were underestimated in the CLR, implying the need to investigate more in depth
the oceanic and atmospheric processes in play in this particular region. As the intraseasonal wind bursts are
related to the fluctuations of St. Helena Anticyclone, their impact on SST variability in the eastern tropical
Atlantic and regional climate suggests the need of better understand the St. Helena Anticyclone variability.
It is also important to note that the mid-May 2005 event occurred during an unusually active year. The year
2005 exhibited a pronounced meridional mode pattern with strong SST gradient between the two hemispheres.
Several authors (Foltz et al., 2006 ; Virmani and Weisberg, 2006 ; Marengo et al., 2008a, 2008b ; Zeng et al.,
2008) studied this particular year, marked by anomalously warm SST in the tropical North Atlantic during
March-July, the warmest from at least 150 years. This anomalous warming was associated with the most active
and destructive hurricane season on record (Foltz et al., 2006; Virmani and Weisberg, 2006) and an extreme and
rare drought in the Amazon Basin (Marengo et al., 2008a, 2008b; Zeng et al. 2008; Erfanian et al., 2017). From
these authors, primary causes of the anomalous warming in 2005 were a weakening of the northeasterly trade
winds and associated decrease in wind-induced latent heat loss as well as changes in shortwave radiation and
horizontal oceanic heat advection. This 2005 temperature record is made even more remarkable given that,
unlike the 1998's one, it occurred in the absence of any strong El Niño anomaly (Shein, 2006). Some studies
(Goldenberg et al., 2001) attribute these SST increases to the Atlantic Multidecadal Oscillation (AMO), while
others suggest that climate change may instead be playing the dominant role (Emanuel, 2005; Webster et al.,
2005; Mann and Emanuel, 2006; Trenberth and Shea, 2006). Comparable anomalously warm tropical Atlantic
SSTs have been observed in 2010 also associated with extreme drought in the Amazon. However, from time
series of monthly anomalies constructed for the two basins (North and South Atlantic) by using OISST monthly
mean data, Erfanian et al. (2017) show that the warmer-than-usual SSTs in the North Atlantic in 2010 was not
associated with colder-than-usual SST in South Atlantic contrarily to 2005 (their Fig. S4e).
While the warming in North Tropical Atlantic during 2005 has been investigated by several authors, the cooling
in South Atlantic has received less attention. This study highlights the need to further document and monitor the
South Atlantic region and the St. Helena Anticyclone, through additional high resolution analysis and
observations.


Acknowledgments:
The research leading to these results received funding from the EU FP7/2007-2013 under grant agreement no.
603521, PREFACE and from the EU Horizon 2020 under grand agreement no. 2014-633211, AtlantOS. These

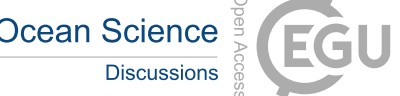



projects are gratefully acknowledged. We do thank Gildas Cambon for his help and participation on the
implementation of ROMS simulations, and Frédéric Marin for his helpful comments.

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
