# Peer review of "Impact of intraseasonal wind bursts on SST variability in the"

_Ocean Science, 2017_

## Referee Comment (RC1) · Anonymous Referee #1 · 16 Nov 2017

This study deals with SST variability in the Cape Lopez region in 2005 and 2006 based on observational data and output of a ROMS model simulation. It is argued that intraseasonal wind bursts from the South Atlantic played an important role for the boreal spring cooling, mainly due to increased upwelling and vertical mixing. A particular focus lies on a cooling episode in May 2005.

The results of the study are interesting and certainly worth publishing in Ocean Science. There are, however, a number of issues that should be clarified. Some of the figures and the writing will need some work as well (see comments below).

[Figure]

Major comments:

1) The study focusses on the years 2005 and 2006. Most of the features discussed in section 4.2, however, appear to occur in both of these year. Maybe it would be more instructive to contrast 2005 to an interannual warm event year?

2) Also, it is not clear whether the processes discussed in section 4 and 5 are specific to 2005 or whether some of them play a role in every spring cooling and/or other inter-annual cold events as well. In other words: Do intraseasonal wind bursts impact SST in the Cape Lopez region in every summer or during every interannual cold event or just in very specific years as 2005?

3) Related to point 1 and 2, the time scales discussed tend to get mixed up a bit. The relationship between the intraseasonal wind bursts, the seasonal cycle of SST, and interannual variations should be sorted out more clearly.

Specific comments:

1) I am missing a motivation on why the Cape Lopez region is of interest.

2) Related to comment (3) above, the time scales of interest should be specified some-where in the beginning, and it should be stated whether the data were filtered or aver-aged over time to focus on them individually.

3) line 184/185: The highest temperatures occur more towards boreal spring than winter.

4) line 188/189: I think all of the references given here discuss biases in coupled climate models while in this study an ocean-only model is used.

5) lines 200-202: A number of previous studies have shown this and could be cited (e.g. Schouten et al., 2005).

6) It is hard to directly compare the conditions in 2005 and 2006 as they are presented in different figures (Fig. 3 and 4) on different pages of the manuscript. I would suggest

to combine those figures. Also, the individual dates given in the text (e.g. lines 231 to 233, lines 257 to 259) are impossible to identify in these figures and should be illustrated in a different way.

7) lines 254/255: Are the data filtered to focus on the intraseasonal time scale? (see comment above)

8) line 336: How did the timing of the preconditioning impact the intensity of the cooling?

9) Fig. 7 and Fig. 10 are very small and thus hard to read.

10) Section 5.2: You mention in the introduction that the monsoon onset happened early in 2005, but this information should be repeated in this section.

11) Fig. 13 looks rather strange because of the discontinuities between May of one and April of the next year. Maybe you could separate the years more clearly with vertical black lines.

12) Instead of Fig. 14 a and b, I would suggest to show a map of the surface pressure for May 2005. The time series can then highlight that the pressure gradient was special.

13) Please check that the figures are numbered in the order in which they are referenced in the text.

Technical corrections/ comments concerning the writing:

line 11: "few documented": please rephrase (e.g. the region of... that has not been studied in detail so far)

line 12, 13 and throughout the text: I would suggest to use the term "cooling episodes" instead of "cooling events". The term "events" sound much longer lasting to me. Alternatively, you could specify the time scale.

line 21: "made the event as a decisive event": please rephrase ("made this event so

special" maybe?)

line 30: "of" instead of "on"

line 69/70: please refer to map shown in Fig. 2

line 88-90: "multiple processes" instead of "multiplicity of processes". I am also not sure what is meant by "very low thermal inertia"

line 99: "trigger" instead of "triggered"

line 103: "show" instead of "put into evidence"

line 189/190: reproduces...very well

line 191: related to

line 223: "Analysis of cooling episodes in the CLR" instead of "Analyze of cooling events in CLR"

line 243/244: "in the CLR" belongs after "cold anomalies"

line 256-259 and thereafter: please refer to figure.

line 320 and 349: "western" instead of "west"

line 364: returned to its initial position

line 439: "generation" instead of "excitation"

line 465: "at play" instead of "in play"

line 533: "in Fig. 12" instead of "on Fig. 12"

line 559: I suggest "What made the mid-May 2005 event so special?"

line 564: "over" instead of "along"

line 628-630: I do not understand that sentence, starting from "but for some"

line 641: an upwelling Kelvin wave

line 647: for the years 2005 and 2006

line 652: "at" instead of "with"

line 669: "that" instead of "than"

Fig. 1: I would suggest to plot the line for 2005 on top of the other lines as it is very hard to see. It would also be helpful to plot a larger area in the maps on the right hand side. What are the vectors shown in Fig. (b) and (c)?

Fig.5: I would suggest to use red for deeper and blue for shallower thermocline to be consistent with SST.

[Figure]

---

## Referee Comment (RC2) · Anonymous Referee #2 · 22 Nov 2017

General Comments

This study uses both models and observations to investigate mechanisms that lead to cold events in the eastern tropical Atlantic Ocean, focusing on the Gulf of Guinea and Cape Lopez region for the period of 1998-2008. The authors then focus on two specific events in the boreal spring of 2005 and 2006, 2005 having the coldest temperatures in the Atlantic cold tongue and 2006 being a normal year. These oceanic regions influence rainfall off the coast of NE Brazil (in the SPCZ) and the West African Monsoon. Results indicate that cold events in this region are brought on by upwelling processes

due to the wind acting on the ocean's surface, which can depend on the depth of the thermocline, and mixing due to vertical shear. The 2005 event was also influenced by a decrease of incoming surface shortwave radiation.

The processes investigated in this paper are very interesting and enhance our knowledge of coupled processes in the South Atlantic; however, I found it a bit difficult to separate the differences of each event both from the text and from the figures. On many of the figures, it is difficult to see the relationships being discussed in the text. Additionally, the authors mention the NE Brazil in the abstract and introduction, but it is hardly mentioned in the analysis, while the West African Monsoon is discussed in the analysis quite a bit but not highlighted in the abstract. This material in this manuscript is worthy of publication. It just needs a bit of work to make it easier to read and understand.

Specific Comments

1. I wonder for many of the plots, especially when discussing the May 2005 event, if it would be better to plot the difference from the climatological mean (an anomaly). It might make the 2005 event stand out. As the figures are, it is difficult to tell that this event is different from some of the other events in the 1998-2005 range.

2. For all figures, it would be helpful to increase the fontsize for the x and y-axis labels. The figures are very difficult to read.

3. It is unclear in the different sections whether the region being discussed is the Cape Lopez region, the equatorial Gulf of Guinea, or the western part of the basin. One confusing discussion revolves around the wind bursts. They are sometimes discussed in the Cape Lopez region associated with southerly winds and sometimes in the western basin as westerly wind bursts associated with Kelvin and Rossby waves. The text mostly just says "wind burst" so it's difficult to tell which is being referenced.

4. On line 13, you say "some particular events iii) a decrease of incoming surface

shortwave radiation," but in fact, you only described one event this applied to (May 2005). This can be fixed by changing the word "some" to "one."

5. Many times in the paper, a season (spring, etc.) is discussed. Please indicate boreal or austral.

6. The paper discusses connections between the South Atlantic and the Cape Lopez region, specifically in relation to the St. Helena Anticyclone. A paper by Bates (J. Clim., 2008) discusses an anomalous low pressure originating in the South Atlantic that migrates northeast-ward, influencing the Southern Trade Winds and thus affecting SST in the Cape Lopez region (though she refers to it as coastal Angola). I don't know if the feature you discuss and the feature she discusses are the same thing. Papers by Bohua Huang and others at the Center for Ocean Land Atmosphere Studies from the 2000s time range also discuss variability in the South Atlantic. You may want to reference these papers if they would add something to your discussion. That is up to the authors to decide.

7. Because you discuss the NE coast of Brazil and the West African Monsoon, it would be nice to have them documented in the seasonal variability section to show how they fit into the normal seasonal cycle.

8. When discussing the thermocline, do you mean shoaling instead of thinning and deepening instead of thickening? You also mention on line 202 that it is at a minimum, I believe you mean "minimum depth."

9. Figure 1 has no scale for the wind speed.

10. I don't think your discussion of Figure 1d on lines 203-205 reflect what is seen in the plot.

11. When you discuss the surface heat flux, please designate whether it is positive downward (into the ocean) or upward (out of the ocean).

12. The individual events mentioned on line 232 are difficult to see. Maybe only plot

April-July or change the y-axis.

13. Lines 275-276: Is the reader supposed to be comparing Fig. 3b with 3d to see the correlation between wind stress and Ekman pumping? If so, it is not clear that this relationship is seen. Also, I don't know how we can see 8degE in this figure. If this correlation is not shown, please say so and let us know what the correlation coefficient is.

14. It might be more telling to try to show the SST/heat content changes in the eastern Atlantic due to each of the processes (upwelling, or even split that into wind stress and vertical mixing, and surface heat fluxes). I'm not sure the best way to suggest this, but perhaps regressions would be suitable. This way, it might be more clear that the May 2005 event was an outlier in terms of short wave cloud radiation.

15. Lines 330-332: I do not see the difference between 2005 and 2006 from Fig. 8. It appears that both Kelvin waves reach the east around the same time and originate in the west around the same time. Figure 6 is also unclear. For 2006, I see many episodes of negative SSH (Feb., Mar., May, June), so why are you only picking the one that occurred in Mar-Apr? I do see a negative value in the east starting a tad earlier in 2006, but not by much. I also see a larger anomaly in 2006 in the east in July-Aug. Why is this not discussed...why only the Mar-Apr event? Is it because you are only focused on the boreal spring event?

16. The text on Fig. 7 is nearly impossible to read.

17. Lines 409-416: This discussion is about southerly wind bursts in the eastern basin, I assume along the coast, but in Fig. 8, I do not see many arrows in that region, so it is difficult to make this connection from the figure.

This paragraph also suggests a linkage between SST variability in the Cape Lopez region and the equatorial region. You might explain this a bit further by discussing the climatological behavior of this connection (like when it occurs and how it develops). I

assume that this is not a feature specific to 2005. I believe that the Bates and Okumura et al. papers might refer to this connection too.

18. Figure 10 is impossible to read, and the features difficult to pick out, especially for the top row and bottom two rows. It would be helpful to mask out the land in all panels and make each panel larger. The text describes a precipitation pattern consistent with a wave train, but I cannot see it because the plot is too small and the arrows seem to be covering the precip pattern.

19. Figure 11: It doesn't seem that you have referred to this figure in the text, though I believe the discussion is on page 21. I do not see what the authors describe in the figure. Perhaps you could be more specific as to the pattern the reader should notice in the plots.

20. Figure 13: It is very difficult to decipher anything from these plots because they are so small and the contour lines are so close together. It is impossible to tell if an event is stronger or not than others. The text says that the 2005 event "appears to be" one of the strongest over the period, but I cannot tell that from this plot. The authors could confirm this by giving the reader a value of wind stress from this period and state that it is confirmed that this is the strongest.

21. Lines 575-577: Is the statement about winds north of the equator relevant to this study? If so, how is this piece of information important?

22. Lines 585-593: Is this relevant to the monsoon discussion? Does the deep convection in the Gulf of Guinea lead to rain and a surface cooling? Is that the impact we should take from this paragraph?

23. Lines 599-602: This paragraph was particularly confusing as to where the wind stress and wind bursts mentioned were located.

24. Lines 716-171: Why exactly does this region need more attention? Because of the effect on the African Monsoon? Please elaborate here to make your conclusion points

better known.

Technical Corrections

There are many English/grammar corrections to be made. I suggest the authors have a native English speaker read through the paper with their input so the meaning isn't lost when correcting the text.

---

## Referee Comment (RC3) · Anonymous Referee #3 · 11 Dec 2017

This manuscript describes conditions in the far eastern equatorial Atlantic during 2005 and 2006. These are interesting years, with strong anomalous cold event in the equatorial Atlantic in 2005 and near-normal conditions in 2006. The main result is that subsurface ocean preconditioning (shoaling of the thermocline through remotely-generated Kelvin waves) and local intraseasonal wind variations caused the strong anomalous cooling event in 2005. A number of processes are proposed to have played important roles, including vertical mixing driven by current shear, surface heat fluxes, and horizontal heat advection by the wind-driven currents and through Rossby wave reflection

at the eastern boundary.

The study is potentially interesting, but is not well organized and is mostly descriptive, with little in-depth analysis. Many topics are discussed briefly, and it's often not clear how they are related to the big picture. Examples are the southward shifts of the ITCZ noted in both years (section 4.2.2b) and precipitation and atmospheric gravity waves (sections 5.1.2 and 5.1.3). It's difficult to follow the main narrative of the manuscript, which I think is the importance of pre-conditioning from equatorial waves and intraseasonal wind-induced mixing and advection and their impacts on SST. It's also unclear how important the chosen region is for local climate and how the changes in that small region are correlated with other indices like the Atlantic cold tongue. As examples of the lack of in-depth analysis, on lines 333-337, preconditioning, local mixing and upwelling, and surface heat flux are mentioned to be important based on brief discussions of equatorial waves, winds, and current shear. This could be quantified better with the model. Similarly, lines 448-450: advection, vertical mixing, and wave propagation are mentioned as factors that extended the SST cooling westward, but no quantification is given. There are many different factors considered, and ultimately it's not clear what is most important. The manuscript would benefit from more in-depth analyses of those mechanisms that are most important and elimination (or reduction) of less important ones.

In general, the figure quality can also be improved significantly. Axis labels are difficult to read. Proper smoothing should be applied to emphasize important time/space scales (this applies to almost all figures). It is also difficult to absorb all the information from the long sequence of map and lon/lat-time contour plot figures. In many cases, the information could be conveyed more clearly and compactly with line plots (possibly Figs. 10-14) of averaged quantities or by combining figures (SSH anomalies with contours of wind anomalies plotted over them).

Other specific comments:

Why focus on this particular region? Is SST in it important for rainfall in a given region?

How are conditions in the CLR related to the cold tongue farther west? What is the correlation between SST in the eastern box and in cold tongue box, for example?

It is difficult to see the differences between Figs. 3 and 4. I suggest replacing with a figure showing differences, or adding a new figure.

How are the results different (or confirm) previous studies of cold tongue variability? It's not clear.

Negative values in Figs. 3c, 4c to me mean shallower than normal thermocline, but it seems you are using the opposite sign so that positive values mean shallower. This is a little confusing. I recommend switching signs or making it clear in the Fig. 3 caption that negative means deeper. Also indicate in the caption that Ekman pumping values >0 indicate upwelling (I assume this is the case?).

Lines 279-292: Do zonal or meridional current variations dominate for the vertical shear, and are they driven by the anomalous meridional winds?

Lines 317-318: What do you mean by "steeper thermocline slope?" Do you mean stronger dT/dz within the thermocline, or shallower thermocline, or stronger horizontal gradients of thermocline depth...

Data/methods section: How are anomalies calculated? It is not stated anywhere, yet shown frequently in the figures. Was the mean seasonal cycle (monthly mean climatology) removed before making Fig. 5, Fig. 6?

I don't see a good correspondence between Figs. 5 and 6. Maybe plotting anomalies from the seasonal cycle would help (if not done already). Otherwise, another method to validate the model's Z20 anomalies is needed.

Line 386: Do you mean Fig. 7c instead of Fig. 6c?

It's difficult to follow the discussion and reasoning on line 380-390. A figure showing spatial patterns of wind anomalies might help to visualize the changes in Ekman pumping and ITCZ shifts.

What is the main result of the analysis discussed on p. 14-15? Why is it important that the southward movement of the ITCZ was more abrupt in 2005 and the winds following the event were different compared to 2006? Please state at the end of the section or mention that it will be discussed in later sections. If it didn't clearly affect later conditions, it should not be shown.

Lines 414-415: How does Fig. 8 show an enhancement of SST cooling after May 10? It only shows SST averaged for May and for May 1-10.

Figure 10: Why not show anomalies for all fields instead of only for winds?

It seems like sections 5.1.2 and 5.1.3 are not essential and could be eliminated.

---

## Author Comment (AC1) · 23 Feb 2018

**Authors' response to Referee 1**

**Journal:**      Ocean Sciences

**Title of paper:**   Impact of intraseasonal wind bursts on SST variability in the far eastern Tropical Atlantic Ocean during boreal spring 2005 and 2006. Focus on the mid-may 2005 event.

**Authors:**      Herbert Gaëlle, Bourlès Bernard.

*We thank Reviewer 1 for his comments and suggestions that allowed improvements of our paper. We have made all needed information to make the figures easily understandable and conforming with general publications criteria (figures size, labels, etc). We also worked to make the manuscript easier to read and understand, by adding some information and removing others.*

**Response to major comments**

**1. RC: The study focusses on the years 2005 and 2006. Most of the features discussed in section 4.2, however, appear to occur in both of these year. Maybe it would be more instructive to contrast 2005 to an interannual warm event year?**

AC: To contrast interannual events in 2005 and in a warm year (like 1998) would be indeed also interesting. However, the comparison with the year 2006, considered as a "normal year", shows that the interannual events are a common feature impacting the SST variability in the studied area and highlights what makes the year 2005 as a "cold" year. To illustrate, the figure X1 (not added in the revised manuscript) below shows the longitude-time diagram of the SST in CLR (Figure X1 a & f) as well as the intraseasonal variations of the wind stress speed (Figure X1 c & h), the 20°C-isotherm depth (Figure X1 d & i), and the sea surface heat flux (Figure X1 e & j) for 1998 (warm year) and 2005 (cold year). We see indeed that the SSTs in boreal spring are higher in spring 1998 than spring 2005. The wind bursts during spring 1998 are not as stronger than during spring 2005. Moreover, the 20°C-isotherm is deeper in spring 1998 than during 2005, making the SST less reactive to wind intensification. What makes the particularity of the year 2005 is not the occurrence of intraseasonal events but their time of occurrence, their strength, and the favorable combination of local and remote

forcing with the arrival of Kelvin wave at the time of strong local winds which induces shallower thermocline.

Thus, we have not described the conditions for 1998 because we have preferred to focus on the year 2005 and understand what makes it an anomalous year compared to a "normal" year. However, it would be interesting to add in the Discussion section some lines about the conditions of a warm year such as 1998. These lines would be added :

"The occurrence of intraseasonal wind intensification in CLR is not specific to the spring/summer 2005 and 2006 and is observed every year over the 1998-2008 period of study (not shown). However, their impact on SST variability in the region is modulated depending on the strength of wind intensification and of the subsurface preconditioning. For example, the year 1998, known as a "warm year", is characterized by anomalous warm SST in boreal spring/summer in the CLR., associated with anomalous weak winds and anomalous deep thermocline. "

[Figure]

Figure X1: (a & f) Latitude-time diagram of 2-daily SST (°C) ; (b & g) intraseasonal variations of SST (°C); (c & h) intraseasonal variations of wind stress; (d & i) 20°C-isotherm depth ; (e & j) surface heat flux; from 1[st] March to 31 August 2005 (left panels) and 1998 (right panels) and averaged from 5°E to 12°E. The intraseasonal variations are computed by remove the 30 days low-pass filtered field to the total field.

**2. RC: Also, it is not clear whether the processes discussed in section 4 and 5 are specific to 2005 or whether some of them play a role in every spring cooling and/or other interannual cold events as well. In other words: Do intraseasonal wind bursts impact**

**SST in the Cape Lopez region in every summer or during every interannual cold event or just in very specific years as 2005?**

AC: The comparison with 2006, considered as a "normal" year, precisely shows that the intraseasonal wind bursts also occur in spring/summer during normal conditions and that is not a particularity of the year 2005. However, in 2005, there are successive strong wind bursts in April-May combined with favorable subsurface conditions (shallow thermocline) due to the arrival of Kelvin wave, that make the cooling more efficient than in 2006 and which occurs earlier than usual. In order to clarify this point in the text, this phrase has been added in the conclusion: "The occurrence of intraseasonal wind intensification in CLR is not specific to the spring/summer 2005 and 2006 and is observed every year over the 1998-2008 period of study (not shown). However, their impact on SST variability in the region is modulated depending on the strength of wind intensification and of the subsurface preconditioning."

**3. RC: Related to point 1 and 2, the time scales discussed tend to get mixed up a bit. The relationship between the intraseasonal wind bursts, the seasonal cycle of SST, and interannual variations should be sorted out more clearly.**

AC: In order to sort out the different times scales more clearly, we decide to show the interannual component of SST/winds/vertical current shear/Ekman pumping variability on figure 3 and 4, by removing the 30-days low-pass filtering to the annual time series. An effort has been made in the text in order to describe more clearly the time scales studied. In addition, some lines have added in section 4.3 ("Westward extension of the CLR cooling") about the climatological behavior of the connection between CLR and equatorial region and the particularity of the year 2005.

Response to Specific comments:

**1. RC: I am missing a motivation on why the Cape Lopez region is of interest.**

AC: The initial reason that motivates the study of the SST variability in the Cape-Lopez region is the observation in satellite SST data of cold coastal waters independent from those observed off shore in the cold tongue region around 10°W (see the map of satellite SST data

for the 8 June 2005 shown on the Figure X2) which raises the question of the link of such cooling with the cold tongue development.

[Figure]

**Figure X2:** SST (°C) from TMI satellite data on 8 June 2005.

The equatorial region and the processes implied in the cold tongue development are largely studied contrary to the Cape-Lopez region. Other several studies focus on SST variability in more southern region such as Angola-Benguela front, but very few in the Cape-Lopez region. However, we thought that better describe the SST variability in the Cape-Lopez region is needed and interesting especially because of the numerous processes in play notably due to the presence of the coast and the proximity of the equator. In addition, some studies (such as DeCoëtlogon et al., 2010) suggest that at short time scale (a few days), more than half of the cold SST anomaly around the equatorial cooling could be explained by horizontal oceanic advection controlled by the winds. Therefore, a better understanding of the SST variability in the CLR may also help to better understand the SST variability in the equatorial region.

**2. RC: Related to comment (3) above, the time scales of interest should be specified somewhere in the beginning, and it should be stated whether the data were filtered or averaged over time to focus on them individually.**

AC: In order to isolate the interannual component, we removed the low-pass filtering (cutoff frequency of 30 days) of the annual time series to the total field. As suggested, we have added this information in the text.

**3. RC: line 184/185: The highest temperatures occur more towards boreal spring than winter.**

AC: Thank you for the remark. Indeed, the highest temperatures occur at the end of March, i.e. at the late of boreal winter and the beginning of boreal spring. The text has been modified accordingly.

**4.RC: line 188/189: I think all of the references given here discuss biases in coupled climate models while in this study an ocean-only model is used.**

AC: Thanks to point this. The phrase line 188/189 has been changed as following:

"Despite a warm bias (~1°C) compared to satellite observations, the model pretty well reproduces the satellite pattern. While this warm bias in the eastern tropical Atlantic is well known in coupled climate models (e.g. Zeng et al., 1996; Davey et al., 2002; Deser et al., 2006; Chang et al., 2007; Richter and Xie, 2008), results from Large and Danabasoglu (2006) show that a warm SST bias may also be present along the Atlantic coast of southern Africa in forced ocean-only simulation."

**5. RC: lines 200-202: A number of previous studies have shown this and could be cited (e.g. Schouten et al., 2005).**

AC: Thank you for point this. Reference has been added as following:

"The region is also characterized by a shallow thermocline which depicts a strong semi-annual cycle (Fig. 1d). The evolution of z20 reveals a thinning of the thermocline during May-July and a thickening up to October-November when it exhibits a minimum, in agreement with previous studies such as the one realized by Schouten et al. (2005) who find a similar seasonal cycle from SSH altimetric data."

**6. RC: It is hard to directly compare the conditions in 2005 and 2006 as they are presented**
**in different figures (Fig. 3 and 4) on different pages of the manuscript. I would suggest to combine those figures. Also, the individual dates given in the text (e.g. lines 231 to 233, lines 257 to 259) are impossible to identify in these figures and should be illustrated in a different way.**

AC: The choice to separate 2005 and 2006 has been made to highlight the correlation between the different fields. In order to have better clarity, we decided to show the total field of SST and 20°C-isotherm depth for 2005 and 2006 on Figure 3 and the intraseasonal variations (by removing the 30-days low-pass filtered data from the total field) of SST/wind/vertical current shear/Ekman pumping for 2005 and 2006 on the same Figure 4, in

order to better highlight the intreaseasonal events. In addition, we have made a zoom on March-August period for better visibility of the events.

[Figure]

**Figure 3:** (a & c) Latitude-time diagram of the sea surface temperature (°C); (b & d) Latitude-time diagram of the 20° C-isotherm depth (m); from 1st March to 31 August 2005 (left panels) and 2006 (right panels) and averaged between 5°E and 12°E.

[Figure]

**Figure 4**: (a & f) Time-latitude diagram, from 7° S to 1° N, of the intraseasonal variations of sea surface temperature (in ° C) averaged between 5° E and 12° E; (b & g) Time evolution of the intraseasonal variations of wind stress amplitude (N.m$^{-2}$) averaged between 5° E and 12° E; (c & h) Latitude-time diagram of the intraseasonal variations of the maximum of the zonal current vertical shear magnitude (m.s$^{-1}$) averaged between 5° E and 12° E ; (d & i) Longitude-time diagram of the intraseasonal variations of Ekman Pumping (m.s$^{-1}$) averaged over the CLR. Ekman pumping values >0 indicate upwelling; (e & j) Latitude-time diagram of the net heat flux (W.m$^{-2}$) averaged between 5° E and 12° E; from 1st March to 31 August 2005 (left panels) and 2006 (right panels).

Modifications have also been made on the plot of 20°C-isotherm depths, Fig.3 : weaker values of 20°C-isotherm depths indicate shallower thermocline to be consistent with the modifications made on the Fig.1, Fig.5, Fig. 9, Fig. 7 and Fig. 13.

**7.RC:  lines 254/255: Are the data filtered to focus on the intraseasonal time scale? (see comment above)**

AC: Yes, the wind stress magnitude field shown on Figure 4 has been obtained after remove the low-pass filtering (cutoff frequency of 30 days) to the total field (see the modified Figure 4 in the response of the previous question).

**8. RC: line 336: How did the timing of the preconditioning impact the intensity of the cooling?**

AC: In 2005, the arrival of the upwelling Kelvin wave in CLR brings the thermocline close to the surface that makes the wind burst, which occurs simultaneously, more efficient in cooling the SST. As explained in line 336, stronger wind intensification and simultaneously favorably preconditioned oceanic subsurface conditions, made the coupling between surface and subsurface ocean processes more efficient than in 2006, resulting in stronger cooling.

**9. RC: Fig. 7 and Fig. 10 are very small and thus hard to read.**

AC: The Figure 7 has been modified and zoomed over January-June. The Figure 10 has been also modified and the wind and precipitation pattern have been separated for more visibility.

[Figure]

**Figure 7**: Time evolution, from 2-days averaged model outputs, of (a) the position (in latitude, between 5° S and 10° N) where the meridional wind stress value equal zero (indicator of the position of the ITCZ), over Jan-June 2005 (left) and Jan-June 2006 (right); (b) the meridional wind stress (N.m$^{-2}$) averaged between 50° W and 35° W and between 1° S and 1° N; (c) same as (b) but for zonal wind stress (N.m$^{-2}$) (in blue); (d) the wind stress curl (N.m$^{-2}$) ; (e) the 20° C isotherm depth (m); (f) the sea level (m). For all fields, except for the position of the ITCZ, the 30 days low-pass filtered annual field averaged over 1998-2008 period has been removed to the total field. The red arrow in (a) indicates the southward shift of the ITCZ before the excitation of the Kevin wave (see text).

[Figure]

**Figure 10**: Daily-averaged, from 13 May to 17 May 2005 (left to right panels), of (a) the precipitation rate (kg.m$^{-2}$/day) ; (b) the wind speed vectors superimposed with wind magnitude (color field) (m.s$^{-1}$) from CFSR fields; (b) the surface pressure (hPa) from ERA-20C reanalysis; (c) the wind speed curl (m.s$^{-1}$) computed from CFSR wind speed fields; and (d) the downward short-wave radiation (W.m$^{-2}$) from CFSR fields.

**10. RC: Section 5.2: You mention in the introduction that the monsoon onset happened early in 2005, but this information should be repeated in this section.**

AC: The following sentences have been added as an introduction of the section 5.2.

: "The mid-May 2005 wind event was found to be involved in the early onset of the ACT development (Marin et al. 2009, Caniaux et al., 2011). Due to the influence of the cold tongue on the WAM onset (Okumura and Xie, 2004; Caniaux et al., 2011; Nguyen et al., 2011; Thorncroft et al., 2011), the mid-May wind event is therefore also linked to the onset of the WAM in 2005 which has been the earliest over 1982-2007 period from Caniaux et al. (2011). In this section we aim to better understand how this single wind event may have such impact."

**11. RC: Fig. 13 looks rather strange because of the discontinuities between May of one and April of the next year. Maybe you could separate the years more clearly with vertical black lines.**

AC: Vertical black thick lines have been added and the figure 13 has been modified for more clarity.

[Figure]

**Figure 13:** Time-latitude diagrams for April-May along the 1998-2008 period, of 2-days average, from top to bottom i) SST (°C); ii) intraseasonal variations anomalies of SST (°C); , iii) intraseasonal variations anomalies of wind stress magnitude (N.m$^{-2}$) from CFSR fields; iv) intraseasonal variations anomalies of short-wave radiation surface flux (W.m$^{-2}$) from CFSR fields; v) intraseasonal variations anomalies of 20°C-isotherm depth (m) computed from the forced model SST; vi) intraseasonal variations anomalies of meridional SST gradient (every 0.5° of latitude), from the forced model; averaged over 10° W-6° W. For all fields, except for the first

SST field, the 30 days low-pass filtered annual field averaged over 1998-2008 period has been removed to the total field.  The vertical black thin line indicates the date of 14 May, 2005.

Modifications have also been made on the plot of 20°C-isotherm depths : weaker values of 20°C-isotherm depths indicate shallower thermocline to be consistent with the modifications made on the Fig.1, Fig.3, Fig.5, Fig. 9 and Fig. 7.

**12. RC: Instead of Fig. 14 a and b, I would suggest to show a map of the surface pressure for May 2005. The time series can then highlight that the pressure gradient was special.**

AC: Thank you for the suggestion. In fact, maps of the surface pressure from May 13 to May 17 2005 are already shown on figure 10. We decided to remove the figure 14 and to modified the comments of the figure 10 about the surface pressure as following:

"The mid-May 2005 event was also characterized by a particularly low surface pressure under the ITCZ, as shown on Fig. 10c. The pressure fall during the mid-May 2005 event appeared as the lowest in May over the whole decade (not shown). It coincided with particularly high surface pressures in St. Helena Anticyclone region 4 days earlier. The meridional surface pressure gradient during the event is thus found to be the strongest over 1998-2008 period. That suggests strong Hadley circulation intensity during the mid-May event and therefore strong anomalous equatorward moisture flux, allowing the deep atmospheric convection in the Gulf of Guinea to be triggered at a self-sustaining level, as previously described in Sect. 5.2**."**

**13. RC: Please check that the figures are numbered in the order in which they are referenced in the text.**

AC: Thanks, this was checked.

**RC: Fig.1: I woud suggest to plot the line for 2005 on top of the other lines as it it very hard to see. It would also be helpful to plot a larger area in the maps on the right hand side. What are the vectors shown in Fig.1b and Fig1c ?**

AC: Thanks for suggestions. The modifications have been made (see Fig.1). The vectors shown in Fig1b and Fig1c are respectively the wind vectors and the surface current vectors. The indications have been added in the legend. In addition,  modifications have been made on the plot of 20°C-isotherm depth: weaker values of 20°C-isotherm depth indicate shallower

thermocline to be consistent with the modifications made on Fig.3, Fig.5, Fig.7, Fig. 9, and Fig. 13.

[Figure]

**Figure 1**: Monthly average of the (a) sea surface temperature (°C); (b) wind stress direction (vectors) and magnitude (color field) (N.m$^{-2}$); (c) horizontal surface current direction (vectors) and speed (color field) (m.s$^{-1}$) ; (d) 20° C-isotherm depth (m); and (e) surface heat flux (W.m$^{-2}$; positive values indicate downward flux) from January to December from 1998 to 2008 and for the climatology (averaged over 1998-2008) simulated by the model (red curve) and from the observations : monthly average TMI 3-daily SST data (light blue curve in (a)); averaged over 5° E-14° E and 7° S-0° S. Right panel: maps of each variable over May-June.

**RC: Fig.5 : I would suggest to use red for deeper and blue for shallower thermocline to be consistent with SST.**

AC: Thanks for suggestion. The modifications have been made on Fig.1, Fig.3, Fig. 5, Fig. 9, Fig. 7, and Fig. 13.

[Figure]

**Figure 5**: Time evolution of the intraseasonal variations anomalies of the 20° C-isotherm depth (m) along the equator (between 54° W and 12° E) and along 9° E (between the equator and 3° S) for 2005 (left) and 2006 (right). Negative values indicate a 20°C isotherm closer to the surface.

[Figure]

**Figure 6:** Time evolution of the sea level anomaly (m) along the equator (between 54° W and 12° E) and along 9° E (between the equator and 3° S) for 2005 (left), and 2006 (right) from AVISO data.

*Additional authors' comments:*

*Thanks a lot for the technical notes. The corrections have been made in the text.*

---

## Author Response (AR1)

**Authors' response to Referee 1**

***Journal:***   Ocean Sciences

***Title of paper:***   Impact of intraseasonal wind bursts on SST variability in the far eastern Tropical Atlantic Ocean during boreal spring 2005 and 2006. Focus on the mid-may 2005 event.

***Authors:***   Herbert Gaëlle, Bourlès Bernard.

*We thank Reviewer 1 for his comments and suggestions that allowed improvements of our paper. We have made all needed information to make the figures easily understandable and conforming with general publications criteria (figures size, labels, etc). We also worked to make the manuscript easier to read and understand, by adding some information and removing others.*

**RC**: Referee's comment; **AC**: Authors' comment; **MC:** Manuscript changes

**Response to major comments**

**1. RC:** **The study focusses on the years 2005 and 2006. Most of the features discussed in section 4.2, however, appear to occur in both of these year. Maybe it would be more instructive to contrast 2005 to an interannual warm event year?**

**AC**: To contrast interannual events in 2005 and in a warm year (like 1998) would be indeed also interesting. However, the comparison with the year 2006, considered as a "normal year", shows that the interannual events are a common feature impacting the SST variability in the studied area and highlights what makes the year 2005 as a "cold" year. To illustrate, the figure X1 (not added in the revised manuscript) below shows the longitude-time diagram of the SST in CLR (Figure X1 a & f) as well as the intraseasonal variations of the wind stress speed (Figure X1 c & h), the 20°C-isotherm depth (Figure X1 d & i), and the sea surface heat flux (Figure X1 e & j) for 1998 (warm year) and 2005 (cold year). We see indeed that the SSTs in boreal spring are higher in spring 1998 than spring 2005. The wind bursts during spring 1998 are not as stronger than during spring 2005. Moreover, the 20°C-isotherm is deeper in spring 1998 than during 2005, making the SST less reactive to wind intensification.

What makes the particularity of the year 2005 is not the occurrence of intraseasonal events but their time of occurrence, their strength, and the favorable combination of local and remote forcing with the arrival of Kelvin wave at the time of strong local winds which induces shallower thermocline.

Thus, we have not described the conditions for 1998 because we have preferred to focus on the year 2005 and understand what makes it an anomalous year compared to a "normal" year. However, it would be interesting to add in the Discussion section some lines about the conditions of a warm year such as 1998.

**MC:** These lines have been added in the conclusion:

*"It should be noted that the occurrence of intraseasonal wind intensification in CLR is not specific to the spring/summer 2005 and 2006 and is observed every year over the 1998-2008 period of study (not shown). However, their impact on SST variability in the region is modulated depending on the strength of wind intensification and of the subsurface preconditioning. For example, the year 1998, known as a "warm year", is characterized by anomalous warm SST in boreal spring/summer in the CLR., associated with anomalous weak winds and anomalous deep thermocline."*

[Figure]

Figure X1: (a & f) Latitude-time diagram of 2-daily SST (°C) ; (b & g) intraseasonal variations of SST (°C); (c & h) intraseasonal variations of wind stress; (d & i) 20°C-isotherm depth ; (e & j) surface heat flux; from 1[st] March to 31 August 2005 (left panels) and 1998 (right panels) and averaged from 5°E to 12°E. The intraseasonal variations are computed by remove the 30 days low-pass filtered field to the total field.

**2. RC: Also, it is not clear whether the processes discussed in section 4 and 5 are specific to 2005 or whether some of them play a role in every spring cooling and/or other interannual cold events as well. In other words: Do intraseasonal wind bursts impact SST in the Cape Lopez region in every summer or during every interannual cold event or just in very specific years as 2005?**

AC: The comparison with 2006, considered as a "normal" year, precisely shows that the intraseasonal wind bursts also occur in spring/summer during normal conditions and that is not a particularity of the year 2005. However, in 2005, there are successive strong wind bursts in April-May combined with favorable subsurface conditions (shallow thermocline) due to the arrival of Kelvin wave, that make the cooling more efficient than in 2006 and which occurs earlier than usual. In order to clarify this point in the text, some lines have been added in the conclusion as mentioned in response to the previous question.

**3. RC: Related to point 1 and 2, the time scales discussed tend to get mixed up a bit. The relationship between the intraseasonal wind bursts, the seasonal cycle of SST, and interannual variations should be sorted out more clearly.**

AC & MC: In order to sort out the different times scales more clearly, we decide to show the interannual component of SST/winds/vertical current shear/Ekman pumping variability on figure 4, by removing the 30-days low-pass filtering to the annual time series. An effort has been made in the text in order to describe more clearly the time scales studied. In addition, some lines have added in section 4.3 ("Westward extension of the CLR cooling") about the climatological behavior of the connection between CLR and equatorial region and the particularity of the year 2005.

**Response to Specific comments:**

**1. RC: I am missing a motivation on why the Cape Lopez region is of interest.**

AC: The initial reason that motivates the study of the SST variability in the Cape-Lopez region is the observation in satellite SST data of cold coastal waters independent from those observed off shore in the cold tongue region around 10°W (see the map of satellite SST data for the 8 June 2005 shown on the Figure X2) which raises the question of the link of such cooling with the cold tongue development.

[Figure]

**Figure X2:** SST (°C) from TMI satellite data on 8 June 2005.

The equatorial region and the processes implied in the cold tongue development are largely studied contrary to the Cape-Lopez region. Other several studies focus on SST variability in more southern region such as Angola-Benguela front, but very few in the Cape-Lopez region. However, we thought that better describe the SST variability in the Cape-Lopez region is needed and interesting especially because of the numerous processes in play notably due to the presence of the coast and the proximity of the equator. In addition, some studies (such as DeCoëtlogon et al., 2010) suggest that at short time scale (a few days), more than half of the cold SST anomaly around the equatorial cooling could be explained by horizontal oceanic advection controlled by the winds. Therefore, a better understanding of the SST variability in the CLR may also help to better understand the SST variability in the equatorial region.

**MC:** Some lines have been added in the Introduction:

*"The question of the processes implied in the SST variability in the Cape-Lopez region was raised based on an observation in satellite SST data of cold coastal waters during spring independent from those observed off shore in the cold tongue region around 10°W which also raised the question of the link of such cooling with the cold tongue development." […] "In addition, some studies (such as DeCoëtlogon et al., 2010) suggest that at short time scale (a few days), more than half of the cold SST anomaly around the equatorial cooling could be explained by horizontal oceanic advection of upwelled cold coastal waters controlled by the winds. Therefore, a better understanding of the SST variability in the CLR may also help to better understand the SST variability in the equatorial region."*

**2. RC: Related to comment (3) above, the time scales of interest should be specified somewhere in the beginning, and it should be stated whether the data were filtered or averaged over time to focus on them individually.**

**AC**: In order to isolate the interannual component, we removed the low-pass filtering (cutoff frequency of 30 days) of the annual time series to the total field.

**MC:** As suggested, we have added this information in the text, in Sect. 2:

*"Note that throughout the whole text and figure captions, the term "intraseasonal variations" is used to designate the field obtained after the removing of the 30 days low-pass filtered field to the total field of the given year, while "intraseasonal anomaly" refers to the field obtained after the removing of the 30 days low-pass filtered field averaged over 1998-2008 to the total field of the given year."*

**3. RC: line 184/185: The highest temperatures occur more towards boreal spring than winter.**

**AC & MC**: Thank you for the remark. Indeed, the highest temperatures occur at the end of March, i.e. at the late of boreal winter and the beginning of boreal spring. The text has been modified accordingly.

**4.RC: line 188/189: I think all of the references given here discuss biases in coupled climate models while in this study an ocean-only model is used.**

**AC**: Thanks to point this.

**MC:** The phrase line 188/189 has been changed as following:

*"Despite a warm bias (~1°C) compared to satellite observations, the model pretty well reproduces the satellite pattern. While this warm bias in the eastern tropical Atlantic is well known in coupled climate models (e.g. Zeng et al., 1996; Davey et al., 2002; Deser et al., 2006; Chang et al., 2007; Richter and Xie, 2008), results from Large and Danabasoglu (2006) show that a warm SST bias may also be present along the Atlantic coast of southern Africa in forced ocean-only simulation."*

**5. RC: lines 200-202: A number of previous studies have shown this and could be cited (e.g. Schouten et al., 2005).**

**AC**: Thank you for point this.

**MC:** Reference has been added as following (section 3):

*"The region is also characterized by a shallow thermocline which depicts a strong semi-annual cycle (Fig. 1d). The evolution of z20 reveals a shoaling of the thermocline during May-July and a deepening up to October-November when it exhibits a maximum depth, in agreement with previous studies such as the one realized by Schouten et al. (2005) who find a similar seasonal cycle from SSH altimetric data."*

**6. RC: It is hard to directly compare the conditions in 2005 and 2006 as they are presented in different figures (Fig. 3 and 4) on different pages of the manuscript. I would suggest to combine those figures. Also, the individual dates given in the text (e.g. lines 231 to 233, lines 257 to 259) are impossible to identify in these figures and should be illustrated in a different way.**

AC: The choice to separate 2005 and 2006 has been made to highlight the correlation between the different fields.

MC: In order to have better clarity, we decided to show the total field of SST and 20°C-isotherm depth for 2005 and 2006 on Figure 3 and the intraseasonal variations (by removing the 30-days low-pass filtered data from the total field) of SST/wind/vertical current shear/Ekman pumping for 2005 and 2006 on the same Figure 4, in order to better highlight the intreaseasonal events. In addition, we have made a zoom on March-August period for better visibility of the events.

[Figure]

**Figure 3:** (a & c) Latitude-time diagram of the sea surface temperature (°C); (b & d) Latitude-time diagram of the 20° C-isotherm depth (m); from 1$^{st}$ March to 31 August 2005 (left panels) and 2006 (right panels) and averaged between 5°E and 12°E.

[Figure]

**Figure 4**: (a & f) Time-latitude diagram, from 7° S to 1° N, of the intraseasonal variations of sea surface temperature (in ° C) averaged between 5° E and 12° E; (b & g) Time evolution of the intraseasonal variations of wind stress amplitude (N.m$^{-2}$) averaged between 5° E and 12° E and between 3° S and 0° S; (c & h) Latitude-time diagram of the intraseasonal variations of the maximum of the current vertical shear magnitude (m.s$^{-1}$) averaged between 5° E and 12°E; (d & i) Longitude-time diagram of the intraseasonal variations of Ekman Pumping (m.s$^{-1}$) averaged over the CLR. Ekman pumping values >0 indicate upwelling; (e & j) Latitude-time diagram of the net heat flux (W.m$^{-2}$) averaged between 5° E and 12° E;  from 1$^{st}$ March to 31 August 2005 (left panels) and 2006 (right panels). For details about calculations of intraseasonal variations, see Sect. 2.

Modifications have also been made on the plot of 20°C-isotherm depths, Fig.3 : weaker values of 20°C-isotherm depths indicate shallower thermocline to be consistent with the modifications made on the Fig.1, Fig.5, Fig. 9, Fig. 7 and Fig. 13 (Fig 12 in revised version).

**7.RC:  lines 254/255: Are the data filtered to focus on the intraseasonal time scale? (see comment above)**

AC: Yes, the wind stress magnitude field shown on Figure 4 has been obtained after remove the low-pass filtering (cutoff frequency of 30 days) to the total field (see the modified Figure 4 in the response of the previous question).

MC: Indications about how the calculations have been made for each figure have been added in the text, in Section 2 : *"Note that throughout the whole text and figure captions, the term "intraseasonal variations" is used to designate the field obtained after the removing of the 30 days low-pass filtered field to the total field of the given year, while "intraseasonal anomaly" refers to the field obtained after the removing of the 30 days low-pass filtered field averaged over 1998-2008 to the total field of the given year."*

**8. RC: line 336: How did the timing of the preconditioning impact the intensity of the cooling?**

AC: In 2005, the arrival of the upwelling Kelvin wave in CLR brings the thermocline close to the surface that makes the wind burst, which occurs simultaneously, more efficient in cooling the SST. As explained in line 336, stronger wind intensification and simultaneously favorably preconditioned oceanic subsurface conditions, made the coupling between surface and subsurface ocean processes more efficient than in 2006, resulting in stronger cooling.

**9. RC: Fig. 7 and Fig. 10 are very small and thus hard to read.**

**AC & MC**: The Figure 7 has been modified and zoomed over January-June. The Figure 10 has been also modified and the wind and precipitation pattern have been separated for more visibility.

[Figure]

**Figure 7:** Time evolution, from 2-days averaged model outputs over Jan-June2005 (left) and Jan-June 2006 (right); of (a & g) the position (in latitude, between 5° S and 10° N) where the meridional wind stress value equal zero (indicator of the position of the ITCZ); (b & h) the intraseasonal anomaly of the meridional wind stress $(N.m^{-2})$ averaged between 50° W and 35° W and between 1° S and 1° N; (c & i) same as (b & h) but for intraseasonal anomaly of zonal wind stress $(N.m^{-2})$; (d & j) the intraseasonal anomaly of the wind stress curl $(N.m^{-2})$ ; (e & k) the intraseasonal anomaly of the 20° C isotherm depth (m); (f & l) the intraseasonal anomaly of the sea level (m). The red arrow in (a & g) indicates the southward shift of the ITCZ before the excitation of the Kevin wave (see text). For details about the calculations of anomalies, see Sect. 2

[Figure]

**Figure 10**: Daily-averaged, from 13 May to 17 May 2005 (left to right panels), of (a) the precipitation rate (kg.m⁻²/day) ; (b) the wind speed vectors superimposed with wind magnitude (color field) (m.s⁻¹) from CFSR fields; (b) the surface pressure (hPa) from ERA-20C reanalysis; (c) the wind speed curl (m.s⁻¹) computed from CFSR wind speed fields; and (d) the downward short-wave radiation (W.m⁻²) from CFSR fields.

**10. RC: Section 5.2: You mention in the introduction that the monsoon onset happened early in 2005, but this information should be repeated in this section.**

**AC & MC:** The following sentences have been added as an introduction of the section 5.2.:
*"The mid-May 2005 wind event was found to be involved in the early onset of the ACT development (Marin et al. 2009, Caniaux et al., 2011). The influence of the cold tongue on the WAM onset has been suggested by several authors (Okumura and Xie, 2004; Caniaux et al., 2011; Nguyen et al., 2011; Thorncroft et al., 2011). At the seasonal time-scale, Caniaux et al. (2011) suggest that it comes from strong interactions between the SST cooling and wind pattern in the eastern equatorial Atlantic: the ACT serves to accelerate (decelerate) winds in the northern (southern) hemisphere contributing to the northward migration of humidify and convection, and pushes precipitation to the continent. Thus, due to its impact on ACT development, the mid-May wind event is also linked to the onset of the WAM in 2005 which has been the earliest over 1982-2007 period from Caniaux et al. (2011). In this section we aim to better understand how this single wind event may have such impact. For further*

*information on the WAM, the reader can refer to Leduc-Leballeur et al. (2013) and Caniaux et al. (2011)."*

**11. RC: Fig. 13 looks rather strange because of the discontinuities between May of one and April of the next year. Maybe you could separate the years more clearly with vertical black lines.**

AC & MC: Vertical black thick lines have been added and the figure 13 has been modified for more clarity.

[Figure]

**Figure 13 ("Figure 12" in the revised manuscript):** Time-latitude diagrams for April-May along the 1998-2008 period, of 2-days average, from top to bottom i) SST (°C); ii) intraseasonal anomaly of SST (°C); , iii) intraseasonal anomaly of wind stress magnitude (N.m$^{-2}$) from CFSR fields; iv) intraseasonal anomaly of short-wave radiation surface flux (W.m$^{-2}$) from CFSR fields; v) intraseasonal anomaly of 20°C-isotherm depth (m) computed from the forced model SST; vi) intraseasonal anomaly of meridional SST gradient (every 0.5° of latitude), from the forced model; averaged over 10° W-6° W. The vertical black thin line indicates the date of 14 May, 2005. For details about the calculations of the anomalies, see Sect. 2.

Modifications have also been made on the plot of 20°C-isotherm depths : weaker values of 20°C-isotherm depths indicate shallower thermocline to be consistent with the modifications made on the Fig.1, Fig.3, Fig.5, Fig. 9 and Fig. 7.

**12. RC: Instead of Fig. 14 a and b, I would suggest to show a map of the surface pressure for May 2005. The time series can then highlight that the pressure gradient was special.**

AC: Thank you for the suggestion. In fact, maps of the surface pressure from May 13 to May 17 2005 are already shown on figure 10.

MC: We decided to remove the figure 14 and to modified the comments of the figure 10 about the surface pressure as following (section 5.1.1) :

*"The strong winds during the event were associated with high pressure core of the Saint Helena Anticyclone, especially on 13-14 May, also associated with particularly low pressure under the ITCZ 4 days later (Fig. 10c). The pressure fall during the mid-May 2005 event appeared as the lowest in May over the whole decade (not shown). The meridional surface pressure gradient during the event is thus found to be the strongest over 1998-2008 period. That suggests strong Hadley circulation intensity during the mid-May event and therefore strong equatorward moisture flux, allowing the deep atmospheric convection in the Gulf of Guinea to be triggered at a self-sustaining level, as previously described in Sect. 5.2."*

**13. RC: Please check that the figures are numbered in the order in which they are referenced in the text.**

AC: Thanks, this was checked.

**RC: Fig.1: I woud suggest to plot the line for 2005 on top of the other lines as it it very hard to see. It would also be helpful to plot a larger area in the maps on the right hand side. What are the vectors shown in Fig.1b and Fig1c ?**

AC & MC: Thanks for suggestions. The modifications have been made (see Fig.1). The vectors shown in Fig1b and Fig1c are respectively the wind vectors and the surface current vectors. The indications have been added in the legend. In addition, modifications have been made on the plot of 20°C-isotherm depth: weaker values of 20°C-isotherm depth indicate shallower thermocline to be consistent with the modifications made on Fig.3, Fig.5, Fig.7, Fig. 9, and Fig. 13 (Fig. 12 in revised version).

[Figure]

**Figure 1**: Monthly average of the (a) sea surface temperature (°C); (b) wind stress direction (vectors) and magnitude (color field) (N.m$^{-2}$); (c) horizontal surface current direction (vectors) and speed (color field) (m.s$^{-1}$) ; (d) 20° C-isotherm depth (m); and (e) surface heat flux (W.m$^{-2}$; positive values indicate downward flux) from January to December from 1998 to 2008 and for the climatology (averaged over 1998-2008) simulated by the model (red curve) and from the observations : monthly average TMI 3-daily SST data (light blue curve in (a)); averaged over 5° E-14° E and 7° S-0° S. Right panel: maps of each variable over May-June.

**RC: Fig.5 : I would suggest to use red for deeper and blue for shallower thermocline to be consistent with SST.**

**AC& MC:** Thanks for suggestion. The modifications have been made on Fig.1, Fig.3, Fig. 5, Fig. 9, Fig. 7, and Fig. 13 (Fig 12 in revised version).

[Figure]

**Figure 5**: Time evolution of the intraseasonal anomaly of 20° C-isotherm depth (m) along the equator (between 54° W and 12° E) and along 9° E (between the equator and 3° S) for 2005 (left) and 2006 (right). Negative values indicate a 20°C isotherm depth closer to the surface. For details about calculations of the anomalies, see Sect. 2.

[Figure]

**Figure 6:** Time evolution of the sea level anomaly (m) along the equator (between 54° W and 12° E) and along 9° E (between the equator and 3° S) for 2005 (left), and 2006 (right) from AVISO data.

*Additional authors' comments:Thanks a lot for the technical notes. The corrections have been made in the text.*

Authors' response to Referee 2

*Journal:*       Ocean Sciences

*Title of paper:*   Impact of intraseasonal wind bursts on SST variability in the far eastern Tropical Atlantic Ocean during boreal spring 2005 and 2006. Focus on the mid-may 2005 event.

*Authors:*       Herbert Gaëlle, Bourlès Bernard.

*We thank Reviewer 2 for his comments and suggestions that allowed improvements of our paper. We have made all needed information to make the figures more understandable and conforming with general publications criteria (figures size, labels, etc). We also worked to make the manuscript easier to read and understand, by adding some information and removing others. The abstract has been also modified taking into account the reviewer's comments (the sentence about the NE Brazil has been removed and some words about the West African Monsoon have been added).*

RC: Referee's comment; AC: Authors' comment; MC: Manuscript changes

**Response to specific Comments**

**1. RC: I wonder for many of the plots, especially when discussing the May 2005 event, if it would be better to plot the difference from the climatological mean (an anomaly). It might make the 2005 event stand out. As the figures are, it is difficult to tell that this event is different from some of the other events in the 1998-2005 range.**

AC: Thanks for the suggestion. Indeed, plot the anomalies allow to better identify the particularity of the mid-May 2005 event.

MC: We have modified the figure 13, enlarged it and the 30-days low-filtered data averaged over 1998-2008 period has been removed to each total field except for the first panel where the SST is shown. In addition, black thick lines have been added to separate each year.

[Figure]

**Figure 13 ("Figure 12" in the revised manuscript):** Time-latitude diagrams for April-May along the 1998-2008 period, of 2-days average, from top to bottom i) SST (°C); ii) intraseasonal anomaly of SST (°C); , iii) intraseasonal anomaly of wind stress magnitude (N.m$^{-2}$) from CFSR fields; iv) intraseasonal anomaly of short-wave radiation surface flux (W.m$^{-2}$) from CFSR fields; v) intraseasonal anomaly of 20°C-isotherm depth (m) computed from the forced model SST; vi) intraseasonal anomaly of meridional SST gradient (every 0.5° of latitude), from the forced model; averaged over 10° W-6° W. The vertical black thin line indicates the date of 14 May, 2005. For details about the calculations of the anomalies, see Sect. 2.

Modifications have also been made on the plot of 20°C-isotherm depths : weaker values of 20°C-isotherm depths indicate shallower thermocline to be consistent with the modifications made on the Fig.1, Fig.3, Fig.5, Fig7 and Fig. 9.

**2. RC: For all figures, it would be helpful to increase the fontsize for the x and y-axis labels. The figures are very difficult to read.**
AC: Thanks for this suggestion. Modifications have been made.

**3. RC: It is unclear in the different sections whether the region being discussed is the Cape Lopez region, the equatorial Gulf of Guinea, or the western part of the basin. One confusing discussion revolves around the wind bursts. They are sometimes discussed in the Cape Lopez region associated with southerly winds and sometimes in the western**

**basin as westerly wind bursts associated with Kelvin and Rossby waves. The text mostly just says "wind burst" so it's difficult to tell which is being referenced.**

**AC**: Thanks for the remark. Indeed, in the first part of the paper we focused on the Cape-Lopez-region and then extend the analysis at more global scale when we focused on the mid-May wind event.

**MC:** For greater clarity, "wind bursts" has replaced by "southerly wind bursts" when they are discussed in the Cape-Lopez region and by "easterly wind bursts" when they are discussed in the western part of the basin.

**4. RC: On line 13, you say "some particular events iii) a decrease of incoming surface shortwave radiation," but in fact, you only described one event this applied to (May 2005). This can be fixed by changing the word "some" to "one."**

**AC**: Thanks for the remark. In fact, another event occurs in spring 2006 (on 2 April).

**MC:** Thus, we included the description of this event in the comments of Figure 4: "*A strong net cooling (-30 W.m$^{-2}$) occurred during the 26-28 May 2005 event. It was mainly due to a sudden decrease of incoming surface short wave radiation (drop of about 80 W.m$^{-2}$ in the CLR between 22 and 28 May; not shown) suggesting increased cloud cover. Another strong net cooling occurred on 2 April 2006 with a mean value in the CLR reaching -95W.m$^2$. It is more sudden than the end-May 2005's one, and was almost exclusively restricted to the CLR region with values reaching locally -185W.m$^2$ (not shown). For both events, the net cooling did not concern the equatorial region west of 0°W.*" Thus, the sentence in the abstract has not been changed.

**5. RC: Many times in the paper, a season (spring, etc.) is discussed. Please indicate boreal or austral.**

**AC & MC**: "spring" has been replaced by "boreal spring".

**6. RC: The paper discusses connections between the South Atlantic and the Cape Lopez region, specifically in relation to the St. Helena Anticyclone. A paper by Bates (J. Clim., 2008) discusses an anomalous low pressure originating in the South Atlantic that migrates northeast-ward, influencing the Southern Trade Winds and thus affecting SST in the Cape Lopez region (though she refers to it as coastal Angola). I don't know if the feature you discuss and the feature she discusses are the same thing. Papers by Bohua Huang and others at the Center for Ocean Land Atmosphere Studies from the**

**2000s time range also discuss variability in the South Atlantic. You may want to reference these papers if they would add something to your discussion. That is up to the authors to decide.**

**AC:** Thanks for this suggestion. Indeed, Bates et al. (2008) show that the patterns of variability in the coastal Angola region is related to fluctuations in the southeast trade winds trough two mechanisms: i) Bjerknes mechanism and ii) variability in subtropical high in South Atlantic. The phenomenon which is at work during May 2005 event related to anomalous strong St Helena Anticyclone, may correspond to the inverse feature that they describe (anomalous low pressure originating in the South Atlantic that migrates northeast-ward, affecting the SST in coastal Angola region with a peaking SST anomalies by approximately 4 months), but at smaller time scale.

**7. RC: Because you discuss the NE coast of Brazil and the West African Monsoon, it would be nice to have them documented in the seasonal variability section to show how they fit into the normal seasonal cycle.**

**AC**: Thanks for the suggestion. However, the NE coast of Brazil is only mentioned in Section 5.1.2 when we describe the anomalous precipitation pattern associated with the mid-May event (early SICZ development linked to the anomalous early development of the equatorial cold tongue). We have thus noted that "This convective zone, located between the ITCZ north of the equator and the South Atlantic Convergence Zone (SACZ) in southern tropics, is the Southern Intertropical Convergence Zone (SICZ) (Grodsky and Carton, 2003). This zone forms usually later, by June-August, when the southern branch of the convection separated from the ITCZ which moves north of the equator."

Thus, it appears to us not necessary to add other information about seasonal variability in this area. More detailed information about 'normal' precipitation conditions in this area can be found in Grodsky and Carton (2003).

About the West African Monsoon, the important point for 2005 is the particularly early onset date, as reported by several authors (such as Caniaux et al. (2011)) associated with the particularly early development of the equatorial cold tongue. The role of the mid-May event in this phenomenon is explained in Section 5.2. We think that it is not necessary to describe more in details the seasonal variations of the West African Monsoon. If the reader needs to have more information about the coastal onset phase of the monsoon in the Gulf of Guinea, he can refer to Leduc-Leballeur et al. (2013), as cited in the text (section 5.2).

**MC:** However, we added these sentences as introduction of the section 5.2:

*"The mid-May 2005 wind event was found to be involved in the early onset of the ACT development (Marin et al. 2009, Caniaux et al., 2011). The influence of the cold tongue on the WAM onset has been suggested by several authors (Okumura and Xie, 2004; Caniaux et al., 2011; Nguyen et al., 2011; Thorncroft et al., 2011). At the seasonal time-scale, Caniaux et al. (2011) suggest that it comes from strong interactions between the SST cooling and wind pattern in the eastern equatorial Atlantic: the ACT serves to accelerate (decelerate) winds in the northern (southern) hemisphere contributing to the northward migration of humidify and convection, and pushes precipitation to the continent. Thus, due to its impact on ACT development, the mid-May 2005 wind event is also linked to the onset of the WAM in 2005 which has been the earliest over 1982-2007 period from Caniaux et al. (2011). In this section we aim to better understand how this single wind event may have such impact. For further information on the WAM, the reader can refer to Leduc-Leballeur et al. (2013) and Caniaux et al. (2011)."*

**8. RC: When discussing the thermocline, do you mean shoaling instead of thinning and deepening instead of thickening? You also mention on line 202 that it is at a minimum, I believe you mean "minimum depth."**

**AC:** Thanks for pointing this. Indeed, when we say "thinning" we mean "shoaling" and when we say "thickening" we mean "deepening". "minimum" is indeed used for "minimum depth". However, following other comments, we modified the sign of z20, therefore, in the modified Fig. 1 the z20 values are positive.

**MC:** Thus, the related sentence has been modified as follows (section 3):

*"The region is also characterized by a shallow thermocline which depicts a strong semi-annual cycle (Fig. 1d). The evolution of z20 reveals a shoaling of the thermocline during May-July and a deepening up to October-November when it exhibits a maximum depth, in agreement with previous studies such as the one realized by Schouten et al. (2005) who find a similar seasonal cycle from SSH altimetric data "*

**9. RC: Figure 1 has no scale for the wind speed.**

**AC:** In fact, the colorbar at the right of the May-June averaged map indicates the scale for the wind stress magnitude.

[Figure]

**Figure 1**: Monthly average of the (a) sea surface temperature (°C); (b) wind stress direction (vectors) and magnitude (color field) (N.m$^{-2}$); (c) horizontal surface current direction (vectors) and speed (color field) (m.s$^{-1}$); (d) 20° C-isotherm depth (m); and (e) surface heat flux (W.m$^{-2}$; positive values indicate downward flux) from January to December from 1998 to 2008 and for the climatology (averaged over 1998-2008) simulated by the model (red curve) and from the observations : monthly average TMI 3-daily SST data (light blue curve in (a)); averaged over 5° E-14° E and 7° S-0° S. Right panel: maps of each variable over May-June.

MC: In addition, modifications have been made on Fig. 1:

- weaker values of 20°C-isotherm depth indicate shallower thermocline to be consistent with the modifications made on Fig.3, Fig.5, Fig.7, Fig. 9, and Fig. 13 (Fig. 12 in revised version).

- May-June averaged maps have been enlarged to better locate the CLR.

**10. RC: I don't think your discussion of Figure 1d on lines 203-205 reflect what is seen in the plot.**

AC: Do you mean "Figure 1e" rather than Figure 1d ? Because the Figure 1d is discussed on lines 200-202 and not on lines 203-205.  For the discussion of Figure 1e, the text has been modified as indicated in our response to the question 8.

**11. RC: When you discuss the surface heat flux, please designate whether it is positive downward (into the ocean) or upward (out of the ocean).**

AC & MC: The sentence "positive values indicate downward flux" has been added in the legend of Fig.1.

**12. RC: The individual events mentioned on line 232 are difficult to see. Maybe only plot April-July or change the y-axis.**

AC: Thanks for the suggestion.

MC: The figures 3 and 4 have been modified in this sense (plot over March-August only). In addition, the intraseasonal variations (removing of the 30 days low-pass filtered field to the total field) of SST/wind stress magnitude/vertical current shear/Ekman pumping are shown on Figure 4 in order to better highlight the intraseasonal events. Modifications have also been made on the plot of 20°C-isotherm depth: weaker values indicate shallower thermocline to be consistent with the modifications made on Fig.1, Fig.5, Fig.7, Fig. 9, and Fig. 13.

[Figure]

**Figure 3:** (a & c) Latitude-time diagram of the sea surface temperature (°C) averaged between 5°E and 12°E; (b & d) Latitude-time diagram of the  20° C-isotherm depth (m) averaged between 5° E and 12° E; from 1st March to 31 August 2005 (left panels) and 2006 (right panels).

[Figure]

**Figure 4**: (a & f) Time-latitude diagram, from 7° S to 1° N, of the intraseasonal variations of sea surface temperature (in ° C) averaged between 5° E and 12° E; (b & g) Time evolution of the intraseasonal variations of wind stress amplitude (N.m$^{-2}$) averaged between 5° E and 12° E and between 3° S and 0° S; (c & h) Latitude-time diagram of the intraseasonal variations of the maximum of the current vertical shear magnitude (m.s$^{-1}$) averaged between 5° E and 12°E; (d & i) Longitude-time diagram of the intraseasonal variations of Ekman Pumping (m.s$^{-1}$) averaged over the CLR. Ekman pumping values >0 indicate upwelling; (e & j) Latitude-time diagram of the net heat flux (W.m$^{-2}$) averaged between 5° E and 12° E; from 1$^{st}$ March to 31 August 2005 (left panels) and 2006 (right panels). For details about calculations of intraseasonal variations, see Sect. 2.

**13. RC: Lines 275-276: Is the reader supposed to be comparing Fig. 3b with 3d to see the correlation between wind stress and Ekman pumping? If so, it is not clear that this relationship is seen. Also, I don't know how we can see 8degE in this figure. If this correlation is not shown, please say so and let us know what the correlation coefficient is.**

**AC & MC**: The Figures 3 and 4 have been modified. We removed the low-pass filtering (cutoff frequency of 30 days) to the total field. The filtered Ekman pumping velocities have been averaged over the area. Thus, the correlation with wind stress is more clearly visible (see the new Figure 3 and 4 in response to the previous question). The text has been modified (section 4.2.1) :

*"The Ekman pumping velocity $w_e$ averaged over the CLR for 2005 and 2006 is shown on Fig. 4d & 4i respectively. The dates of intraseasonal upward velocities are quite well correlated with the dates of intraseasonal wind events (with correlation coefficient equal to 0.55 for 2005 and 0.41 for 2006),*

*maximum being during the early-April, mid-May and end-May 2005 events and during late April, mid-June and end-June 2006. However, for comparable wind intensification, the boreal spring and summer wind events were not associated with comparable intensity of Ekman pumping velocity."*

**14. RC: It might be more telling to try to show the SST/heat content changes in the eastern Atlantic due to each of the processes (upwelling, or even split that into wind stress and vertical mixing, and surface heat fluxes). I'm not sure the best way to suggest this, but perhaps regressions would be suitable. This way, it might be more clear that the May 2005 event was an outlier in terms of short wave cloud radiation.**

AC: Thanks for the suggestion. Showing the heat content changes in the eastern Atlantic due to each of the processes would be indeed interesting. However, we consider that showing the Ekman pumping, vertical current shear, and surface heat flux bring the relevant information needed to explain the main processes at play.

MC: However, in order to better highlight the particularity of each wind event, we have modified the figure, zoomed from 1$^{st}$ March to 31 August 2005 and 2006 and shown the intraseasonal variations for SST, wind, Ekman pumping, and vertical current shear. The net surface heat flux have been not filtered in order to highlight the events characterized by negative heat flux, such as the end-May 2005 event and the beginning of April 2006 event.

**15. RC: Lines 330-332: I do not see the difference between 2005 and 2006 from Fig. 8. It appears that both Kelvin waves reach the east around the same time and originate in the west around the same time. Figure 6 is also unclear. For 2006, I see many episodes of negative SSH (Feb., Mar., May, June), so why are you only picking the one that occurred in Mar-Apr? I do see a negative value in the east starting a tad earlier in 2006, but not by much. I also see a larger anomaly in 2006 in the east in July-Aug. Why is this not discussed…why only the Mar-Apr event? Is it because you are only focused on the boreal spring event?**

AC: Do you mean « Fig. 5 » instead of "Fig. 8" ?

On Fig.5 and 6, discussed on lines 330-332, the Kelvin waves in 2005 and 2006 are delayed by about 15 days. Even weak, such a 15 days difference contributes to make the thermocline shallower when the mid-May wind burst occurs in 2005. However, it is true that the difference is not so easy to observe from Figure 5 & 6.

MC: Therefore, for more clarity, the sentence on line 330-332 (section 4.2.2.a) has been modified as follows:

*"In 2005, negative SSH and z20 anomalies occurred in the West in early March-early April and in mid-May, whereas they occurred around late-February – mid-March and early May and June in 2006. The first Kelvin wave thus reached the CLR slightly earlier in 2006 than 2005, at the beginning of May. In addition, the two upwelling Kelvin waves followed each other more closely in 2005 than in 2006."*

Moreover, the figures have been modified and we have plotted the anomalies only for the period March-August for better clarity. We focus on negative SSH occurred in Mar-Apr in the west because that is this event which induces a shallower thermocline in the east few weeks later, in April-May. Indeed, we focused on the boreal spring events in the east.

[Figure]

**Figure 5**: Time evolution of the intraseasonal anomaly of 20° C-isotherm depth (m) along the equator (between 54° W and 12° E) and along 9° E (between the equator and 3° S) for 2005 (left) and 2006 (right). Negative values indicate a 20°C isotherm depth closer to the surface. For details about calculations of the anomalies, see Sect. 2.

[Figure]

**Figure 6:** Time evolution of the sea level anomaly (m) along the equator (between 54° W and 12° E) and along 9° E (between the equator and 3° S) for 2005 (left), and 2006 (right) from AVISO data.

**16. RC: The text on Fig. 7 is nearly impossible to read.**

**AC & MC:** Sorry for that. Modifications have been made on the figure 7 for more clarity.

[Figure]

**Figure 7**: Time evolution, from 2-days averaged model outputs over Jan-June2005 (left) and Jan-June 2006 (right); of (a & g) the position (in latitude, between 5° S and 10° N) where the meridional wind stress value equal zero (indicator of the position of the ITCZ); (b & h) the intraseasonal anomaly of the meridional wind stress (N.m$^{-2}$) averaged between 50° W and 35° W and between 1° S and 1° N; (c & i) same as (b & h) but for intraseasonal anomaly of zonal wind stress (N.m$^{-2}$); (d & j) the intraseasonal anomaly of the wind stress curl

(N.m$^{-2}$) ; (e & k) the intraseasonal anomaly of the 20° C isotherm depth (m); (f & l) the intraseasonal anomaly of the sea level (m). The red arrow in (a & g) indicates the southward shift of the ITCZ before the excitation of the Kevin wave (see text). For details about the calculations of anomalies, see Sect. 2.

**17. RC: Lines 409-416: This discussion is about southerly wind bursts in the eastern basin, I assume along the coast, but in Fig. 8, I do not see many arrows in that region, so it is difficult to make this connection from the figure.**
**This paragraph also suggests a linkage between SST variability in the Cape Lopez region and the equatorial region. You might explain this a bit further by discussing the climatological behavior of this connection (like when it occurs and how it develops). I assume that this is not a feature specific to 2005. I believe that the Bates and Okumura et al. papers might refer to this connection too.**

AC & MC: The figure 8 has been modified for better visibility.

[Figure]

**Figure 8**: (a) intraseasonal anomaly of sea surface temperature (° C; color) superimposed with intraseasonal anomaly of wind stress intensity (arrows) averaged over 1-12 May 2005 (up panel) and over 14-30May (down panel); (b) same but for 2006. For details about the calculations of the anomalies, see Sect.2.

Indeed, the connection between the Cape-Lopez region around 3°S and the southern edge of the equatorial cold tongue is not specific to 2005 and 2006. The westward extension of the cold SST takes place every year over 1998-2008 period (our period of study) but starts at different time. It occurs generally from June-July, when the cooling events usually occur in the east at this location, and is thus closely linked with the shoaling of the thermocline due to the arrival of the Kelvin upwelling wave at the coast. In 2005, the strongest cooling events induced by strong southerly winds occur earlier, in May, combined with anomalous shallower thermocline due to early arrival of Kelvin upwelling wave.  The cooling in the CLR also reaches more coastal area due to anomalous strong wind events in the east part of the basin

while it does not reach the coast at this location (3°S) in boreal spring for the most years over 1998-2008 period. In addition, the westward surface currents are usually maximum in boreal spring (as visible on the seasonal cycle shown on Fig.1) and extend over the most coastal area in the east during southerly wind events. They can thus even more contribute to the westward extension of cold coastal waters in May 2005.

In 2006, the westward extension of cold waters established from the beginning of July. Yet, coastal cooling occurred at the end of May but no westward extension of the cold waters is observed at this period. In 2005, the two upwelling Kelvin wave followed each other closely while in 2006, the first Kelvin upwelling wave reaches the coast in May and the second in July. In addition, the wind event responsible of the cooling at the end of May 2006 is rather isolated and less strong than the one in mid-May 2005 (which is preceded and followed by another wind bursts few days before and after). In order to clarify these points in the paper, we added a figure for the year 2006 and modified the comments in the text as follows:

[revised manuscript text omitted]

**18. RC: Figure 10 is impossible to read, and the features difficult to pick out, especially for the top row and bottom two rows. It would be helpful to mask out the land in all panels and make each panel larger. The text describes a precipitation pattern consistent with a wave train, but I cannot see it because the plot is too small and the arrows seem to be covering the precip pattern.**

AC & MC: The figure 10 has been modified. The precipitation and wind patterns have been separated and the plots enlarged.

[Figure]

Figure 10: Daily-averaged, from 13 May to 17 May 2005 (left to right panels), of (a) wind magnitude (color field) (m.s⁻¹) superimposed with wind vectors from CFSR fields; (b) precipitation rate (kg.m⁻²/day) ⁻¹) from CFSR fields; (b) surface pressure (hPa) from ERA-20C reanalysis; (c) wind speed curl (m.s⁻¹) computed from CFSR wind speed fields; and (d) downward short-wave radiation (W.m⁻²) from CFSR fields.

**19. RC: Figure 11: It doesn't seem that you have referred to this figure in the text, though I believe the discussion is on page 21. I do not see what the authors describe in the figure. Perhaps you could be more specific as to the pattern the reader should notice in the plots**.

**AC & MC**: We decided to remove the figure 11. The text has thus been modified as follows:

*"The precipitation fields during the mid-May event (Fig. 10a) also evidence rainfall pattern typical of atmospheric gravity wave train characterized by a horizontal wave length ~500 km and initiated by a front system (forming the northern boundary of a low pressure system) which developed around 17° S on 14 May and traveled northeastward until 17 May. The rainfall train was associated with oscillatory wind stress curl train alternating between positive and negative anomalies (Fig. 10c) as well as alternating downward shortwave radiation minimum (Fig. 10d) associated with the wave clouds. Gravity waves are known to play an important role in transporting the momentum and energy through long distances (Fritts, 1984). Here, they would be a way to carry momentum and energy from South Atlantic to the equator during the strong event."*

**20. RC: Figure 13: It is very difficult to decipher anything from these plots because they are so small and the contour lines are so close together. It is impossible to tell if an event is stronger or not than others. The text says that the 2005 event "appears to be" one of the strongest over the period, but I cannot tell that from this plot. The authors could confirm this by giving the reader a value of wind stress from this period and state that it is confirmed that this is the strongest.**

**AC & MC**: The figure 13 has been modified for more clarity: vertical black lines have been added to separate the years and the value of wind stress anomaly during the 2005 event has been added in the text (up to 0.13N.m² around 15°S and 0.05N.m² in equatorial region). In addition, we decided to show the fields after removing the 30 days-low pass filtered field averaged over 1998-2008 period, except for the first panel which shows the SST total field.

[Figure]

**Figure 13 ("Figure 12" in the revised manuscript"):** Time-latitude diagrams for April-May along the 1998-2008 period, of 2-days average, from top to bottom i) SST (°C); ii) intraseasonal anomaly of SST (°C); , iii) intraseasonal anomaly of wind stress magnitude (N.m$^{-2}$) from CFSR fields; iv) intraseasonal anomaly of short-wave radiation surface flux (W.m$^{-2}$) from CFSR fields; v) intraseasonal anomaly of 20°C-isotherm depth (m) computed from the forced model SST; vi) intraseasonal anomaly of meridional SST gradient (every 0.5° of latitude), from the forced model; averaged over 10° W-6° W. The vertical black thin line indicates the date of 14 May, 2005. For details about the calculations of the anomalies, see Sect. 2.

Modifications have also been made on the plot of 20°C-isotherm depths : weaker values of 20°C-isotherm depths indicate shallower thermocline to be consistent with the modifications made on the Fig.1, Fig.3, Fig.5, Fig. 9, and Fig. 7.

**21. RC: Lines 575-577: Is the statement about winds north of the equator relevant to this study? If so, how is this piece of information important?**

AC: The wind-strengthening events north of the Equator during boreal spring in the Gulf of Guinea is implied in the rainfall coastal onset and is linked to the intraseasonal southerly wind burst. Indeed, from Leduc Leballeur et al. (2013), the enhancement and maintenance of southerly winds north of the equator in the Gulf of Guinea is linked to a coincident installation of a deep circulation and a northward shift of the low atmospheric local circulation. This wind strengthening on the northern side of the Equator contributes to the

northward migration of humidity and convection, and pushes precipitation to the continent. It is an indication of the "rainfall coastal onset" of the monsoon.

In section 4.2, we show that as of date of the mid-May 2005 event, the wind north of the equator becomes and remains strong indicating that the mid-May 2005 event is the trigger event of the rainfall coastal onset. The strengthening of winds north of the equator is due to the meridional SST gradient created at the equator during the event. The figure 13 (Figure 12 in revised manuscript) shows that the meridional SST gradient during May 2005 is indeed anomalous strong compared to April-May usual conditions.  That what we noted by the sentence in Sect. 5.3: *"This meridional SST gradient was responsible for the wind surface intensification north of the equator (Fig. 11a and Fig. 12, fourth panel) through air-sea interaction mechanisms as described by Leduc-Leballeur et al. (2011)."*

**22. RC: Lines 585-593: Is this relevant to the monsoon discussion? Does the deep convection in the Gulf of Guinea lead to rain and a surface cooling? Is that the impact we should take from this paragraph?**

AC: The wind strengthening results in equatorial surface cooling, which in turns intensifies the southerlies north of the Equator through air-sea interaction. This increases convection in the northern Gulf of Guinea, accompanied with a northward shift of the precipitation. Generally, in May the low atmospheric local circulation (LALC) appears briefly due to southeastern wind burst and collapses within a few days. The establishment of the LALC at a self-sustaining level appears usually at the end of May-beginning of June, triggered by a significantly stronger southeasterly wind burst. We show that in 2005, the mid-May event is this significantly stronger southeastern wind burst. It is especially particular because it appears 15 days before the averaged reference date computed by Leduc-Leballeur et al. (2011)                   over                   the                   2000-2009                   period.

MC: The paragraph on lines 585-593 and the figure 14 have been deleted, and the high pressure in St Helena anticyclone region and the low pressures in Gulf of Guinea  are now shown on figure 10, section 5.1.1. Moreover, we have deleted the comments about the pressure gradient in section 5.3 and added the lines below in section 5.1.1: *"The strong winds during the event were associated with high pressure core of the Saint Helena Anticyclone, especially on 13-14 May, also associated with particularly low pressure under the ITCZ 4 days later (Fig. 10c). The pressure fall during the mid-May 2005 event appeared as the lowest in May over the whole decade (not shown). The meridional surface pressure gradient during the event is thus found to be the strongest over 1998-2008 period. That suggests strong Hadley circulation intensity during the mid-*

*May event and therefore strong equatorward moisture flux, allowing the deep atmospheric convection in the Gulf of Guinea to be triggered at a self-sustaining level, as previously described in Sect. 5.2.*"

.**23. RC: Lines 599-602: This paragraph was particularly confusing as to where the wind stress and wind bursts mentioned were located.**

AC: The wind burst mentioned lines 599-602 is the one evidenced on figure 13 (Figure 12 in revised version) during the year 2000, over 10°W-6°W region.

MC: The sentence on line 599-600 has been modified as follows (Section 5.3): *"Another southerly wind burst of comparable intensity occurred at the beginning of May 2000 (Fig. 12, fourth panel) while the thermocline was shallow, causing SST cooling at the equator (Fig. 12, first and second panels)."*

**24. RC: Lines 716-171: Why exactly does this region need more attention? Because of the effect on the African Monsoon? Please elaborate here to make your conclusion points better known.**

AC: The South Atlantic region, and in particular the St. Helena Anticyclone variability, need more attention because of the impact of its fluctuations on the SST variability in the tropical Atlantic and in particular on the equatorial cold tongue development, as showed in the paper. The energy from South Atlantic is indeed carried toward lower latitudes by different ways : i) direct effect of the southerly winds in the east, ii) energy transport via atmospheric gravity waves, iii) excitation of Kelvin wave in the West by southeasterly winds.

In our paper, we show that intraseasonal wind bursts, related to St Helena Anticyclone fluctuations have an impact on SST variability in the CLR generating cold events in boreal spring/summer. Other studies, as the one realized by Marin et al. (2009) showed that they also impact the SST variability in the cold tongue region. In addition, the influence of the cold tongue on West African monsoon onset has been suggested by many authors (e.g. Okumura and Xie, 2004; Caniaux et al., 2011; Nguyen et al., 2011; Thorncroft et al., 2011). In 2005, we show that a particularly strong wind burst is responsible for a particularly early coastal monsoon onset. Thus, a better understanding of the variability of St. Helena Anticylone at intraseasonal time scales would allow to bring further information about these processes.

*In addition to modifications listed above, many English/grammar corrections have been made in the text.*

Authors' response to Referee 3

*Journal:*        Ocean Sciences

*Title of paper:*    Impact of intraseasonal wind bursts on SST variability in the far eastern Tropical Atlantic Ocean during boreal spring 2005 and 2006. Focus on the mid-may 2005 event.

*Authors:*       Herbert Gaëlle, Bourlès Bernard.

*We thank Reviewer 3 for his comments and suggestions that allowed improvements of our paper. We have made all needed modifications to make the figures easily understandable and conforming with general publications criteria (figures size, labels, etc). We have also made effort to make the main narrative of the manuscript easier to follow. A more in-depth analysis would have been obviously interesting but we first aimed to understand the different processes acting in the region. In addition, a more in-depth analysis of one or two particular processes would have prevented the description of the succession of the processes as a whole.*

**RC**: Referee's comment; **AC**: Authors' comment; **MC:** Manuscript changes

**Response to specific comments:**

**RC: Why focus on this particular region? Is SST in it important for rainfall in a given region?**

**AC**: The initial reason that motivates the study of the SST variability in the Cape-Lopez region is the observation in satellite SST data of cold coastal waters independent from those observed off shore in the cold tongue region around 10°W (see the map of satellite SST data for the 8 June 2005 shown on the Figure X1) which raises the question of the link of such cooling with the cold tongue development.

[Figure]

**Figure X1:** SST (°C) from TMI satellite data on 8 June 2005.

The equatorial region and the processes implied in the cold tongue development are largely studied contrary to the Cape-Lopez region. Other several studies focus on SST variability in more southern region such as Angola-Benguela front, but very few in the Cape-Lopez region. However, we thought that better describe the SST variability in the Cape-Lopez region is needed and interesting especially because of the numerous processes in play notably due to the presence of the coast and the proximity of the equator. In addition, some studies (such as DeCoëtlogon et al., 2010) suggest that at short time scale (a few days), more than half of the cold SST anomaly around the equatorial cooling could be explained by horizontal oceanic advection controlled by the winds. Therefore, a better understanding of the SST variability in CLR may also help to better understand the SST variability in equatorial region.

**MC:** Some lines have been added in the Introduction:

*"The question of the processes implied in the SST variability in the Cape-Lopez region was raised based on an observation in satellite SST data of cold coastal waters during spring independent from those observed off shore in the cold tongue region around 10°W which also raised the question of the link of such cooling with the cold tongue development." […] "In addition, some studies (such as DeCoëtlogon et al., 2010) suggest that at short time scale (a few days), more than half of the cold SST anomaly around the equatorial cooling could be explained by horizontal oceanic advection of upwelled cold coastal waters controlled by the winds. Therefore, a better understanding of the SST variability in the CLR may also help to better understand the SST variability in the equatorial region."*

**RC: How are conditions in the CLR related to the cold tongue farther west? What is the correlation between SST in the eastern box and in cold tongue box, for example?**

**AC**: Given that the CLR and cold tongue region are submitted to the similar atmospheric forcing, the SST variability in both regions is quite close (cooling event at the same date). However, the processes responsible of the cooling differ from CLR region to cold tongue region due in particular to the presence of the coast. From many authors (Yu et al., 2006; Peter et al., 2006; Wade et al., 2011; Jouanno et al., 2011), the cooling in the cold tongue region would be regulated by a coupling between thermocline shoaling and subsurface dynamics such as turbulent mixing, vertical advection and entrainment, as well as horizontal advection.

In the CLR, we showed that upwelling processes are involved in particular around 3-4°S, as well as vertical current shear implying the SEC, which is enhanced during southerly wind bursts. Our analysis for the year 2005 and 2006 has also shown that during particular events (at the end of May and beginning of April 2006), a decrease of short wave radiation in CLR due to increased cloud cover contributes to the cooling. This phenomenon doest not concern the equatorial region east of 0°W. In addition, for a given wind burst, the intensity of SST response in CLR and cold tongue region will modulate by subsurface conditions which are under the influence of equatorial Kelvin wave. For example in May 2005, the Kelvin wave reached the eastern coast while three wind bursts occurred, thus the thermocline was shallower in the east than west of 0°W. We also highlighted westward extension of cold eastern upwelled water around 3°S through combined effects of westward surface currents, local wind influences and wave westward propagation which may contribute to the cooling in the southern edge of the cold tongue region.

**MC:** Some lines about this have been added at the end of the section 4:

*"In conclusion to this section 4, the SST variability in the CLR at intraseasonal time scales is the result of combination between basin preconditioning by remotely forced shoaling of the thermocline via Kelvin wave, local mixing induced by current vertical shear, and upwelling processes in response to strong southerly winds. As highlighted for the 26-28 May 2005 and 2 April 2006 events, the net heat flux may also contribute to cool the surface waters, through enhanced cloud cover which decrease the incoming solar radiation. The cold upwelled waters around 3°S extend then westward from the eastern coast to near 20°W by combined effect of the westward propagating Rossby waves as well as vertical mixing and advection processes. The cool water may thus contribute to the cooling in the southern edge of the cold tongue region. Although the processes implied differ slightly due to the presence of the coast, the SST variability in the CLR is quite close to the one in the equatorial cold tongue region (not shown), due to similar atmospheric forcing. However, for a given wind burst, the intensity of SST response in the CLR and in the cold tongue region is modulated by subsurface conditions which are under the influence of equatorial Kelvin wave. In May 2005, the Kelvin wave*

*reached the eastern coast while three wind bursts occurred. The thermocline was thus shallower in the east than west of 0°W, providing favorable subsurface conditions making the coupling between making the SST more reactive to wind intensification occurred during this month. In addition, the decrease short wave radiations due to enhanced cloud cover during the 26-28 May 2005 event or 2 April 2006 event, which contribute to the cooling in the CLR, did not concern the equatorial region east of 0°W."*

**RC: It is difficult to see the differences between Figs. 3 and 4. I suggest replacing with a figure showing differences, or adding a new figure.**

AC & MC: The figures 3 and 4 have been modified. The filtered SST (where the 30days-low pass filtered field has been removed to the total field) has been added in order to better highlight the cold episodes. In addition, a zoom over March-August period has been made for better clarity.

[Figure]

**Figure 3:** (a & c) Latitude-time diagram of the sea surface temperature (°C) averaged between 5°E and 12°E; (b & d) Latitude-time diagram of the 20° C-isotherm depth (m) averaged between 5° E and 12° E; from 1st March to 31 August 2005 (left panels) and 2006 (right panels).

[Figure]

**Figure 4**: (a & f) Time-latitude diagram, from 7° S to 1° N, of the intraseasonal variations of sea surface temperature (in ° C) averaged between 5° E and 12° E; (b & g) Time evolution of the intraseasonal variations of wind stress amplitude (N.m$^{-2}$) averaged between 5° E and 12° E and between 3° S and 0° S; (c & h) Latitude-time diagram of the intraseasonal variations of the maximum of the current vertical shear magnitude (m.s$^{-1}$) averaged between 5° E and 12°E; (d & i) Longitude-time diagram of the intraseasonal variations of Ekman Pumping (m.s$^{-1}$) averaged over the CLR. Ekman pumping values >0 indicate upwelling; (e & j) Latitude-time diagram of the net heat flux (W.m$^{-2}$) averaged between 5° E and 12° E; from 1$^{st}$ March to 31 August 2005 (left panels) and 2006 (right panels). For details about calculations of intraseasonal variations, see Sect. 2.

Modifications have also been made on the plot of 20°C-isotherm depths : weaker values of 20°C-isotherm depths indicate shallower thermocline to be consistent with the modifications made on the Fig.1, Fig.5, Fig.7, Fig. 9 and Fig. 13 (Fig. 12 in revised version) in response to the other reviewers' comments.

**RC: How are the results different (or confirm) previous studies of cold tongue variability? It's not clear.**

**AC:** Our study does not focus on the cold tongue variability but, first, on SST variability more eastern, in the Cape-Lopez region. Marin et al. (2009) show that the cooling in 10°W-4°W region is the result of successive cooling events related to intraseasonal wind bursts. The two regions are under the influence of similar atmospheric forcing but the processes implied are rather different. We show that the SST in the CLR also reacts to the intraseasonal wind

bursts. However, the processes responsible of the cooling differ from the CLR region to the cold tongue region due in particular to the presence of the coast (see our response to the previous question "How are conditions in the CLR related to the cold tongue farther west? What is the correlation between SST in the eastern box and in cold tongue box, for example?**").

The cold tongue region is mentioned in the second part of our paper when we focus on the mid-May 2005 wind burst and its impact on coastal monsoon onset. Indeed, we aim to describe the wind burst impacting the Cape-Lopez region at more global scale, so we analyzed its impact in the Cape-Lopez region and also in the cold tongue region through its role in West African Monsoon onset.

**RC: Negative values in Figs. 3c, 4c to me mean shallower than normal thermocline, but it seems you are using the opposite sign so that positive values mean shallower. This is a little confusing. I recommend switching signs or making it clear in the Fig. 3 caption that negative means deeper. Also indicate in the caption that Ekman pumping values >0 indicate upwelling (I assume this is the case?).**

**AC & MC**: Thanks for this suggestion. We have modified the figures 3c and 4c in this sense and we have added that Ekman pumping values >0 indicate upwelling in the captions of the figures.

**RC: Lines 279-292: Do zonal or meridional current variations dominate for the vertical shear, and are they driven by the anomalous meridional winds?**

**AC & MC**: The vertical shear is dominated by zonal current variations, related to the fluctuations of dominant southerly winds. We have modified the figure 3 and 4 where we plotted the vertical shear magnitude (see the response to the previous comment: "It is difficult to see the differences between Figs. 3 and 4. I suggest replacing with a figure showing differences, **or** adding a new figure."). On the new figures, we also removed the 30-days low-pas filtered field to the total field.

**RC: Lines 317-318: What do you mean by "steeper thermocline slope?" Do you mean stronger dT/dz within the thermocline, or shallower thermocline, or stronger horizontal gradients of thermocline depth...**

**AC**: By 'steeper thermocline slope' we mean 'shallower thermocline".

**MC:** We have clarified this in the text.

**RC: Data/methods section: How are anomalies calculated? It is not stated anywhere, yet shown frequently in the figures. Was the mean seasonal cycle (monthly mean clima-tology) removed before making Fig. 5, Fig. 6?**

**AC:** For the Figure 5, we applied a 30-days low-pass filter to the total field, averaged the result over 1998-2008 period and removed it to the total field of each year.

**MC:** Indications about how the calculations have been made for each figure have been added in the text, in Section 2 *: "Note that throughout the whole text and figure captions, the term "intraseasonal variations" is used to designate the field obtained after the removing of the 30 days low-pass filtered field to the total field of the given year, while "intraseasonal anomaly" refers to the field obtained after the removing of the 30 days low-pass filtered field averaged over 1998-2008 to the total field of the given year."*

**RC: I don't see a good correspondence between Figs. 5 and 6. Maybe plotting anomalies from the seasonal cycle would help (if not done already). Otherwise, another method to validate the model's Z20 anomalies is needed.**

**AC**: The figures 5 and 6 have been modified. Negative values of 20°C-isotherm depth now show shallower thermocline, to better highlight the correspondence between Fig. 5 and Fig. 6. The values plotted on Fig.5 are obtained by removing the 30-days low pass filtered field, averaged over 1998-2008 period to the total field.

[Figure]

**Figure 5**: Time evolution of the intraseasonal anomaly of the 20° C-isotherm depth (m) along the equator (between 54° W and 12° E) and along 9° E (between the equator and 3° S) for 2005 (left) and 2006 (right). Negative values indicate a 20°C isotherm closer to the surface.

[Figure]

**Figure 6:** Time evolution of the sea level anomaly (m) along the equator (between 54° W and 12° E) and along 9° E (between the equator and 3° S) for 2005 (left), and 2006 (right) from AVISO data. For details about calculations of the anomalies, see Sect. 2.

**RC: Line 386: Do you mean Fig. 7c instead of Fig. 6c?**

**It's difficult to follow the discussion and reasoning on line 380-390. A figure showing spatial patterns of wind anomalies might help to visualize the changes in Ekman pumping and ITCZ shifts.**

AC: Yes, sorry for that, we indeed mean Fig. 7c instead of Fig. 6c. We are not sure that it is necessary to add additional figure. We think that what we want to show is clearly visible on the plot of Fig. 7.

**RC: What is the main result of the analysis discussed on p. 14-15? Why is it important that the southward movement of the ITCZ was more abrupt in 2005 and the winds following the event were different compared to 2006? Please state at the end of the section or mention that it will be discussed in later sections. If it didn't clearly affect later conditions, it should not be shown.**

AC: In the previous section (4.2), we show that the intraseasonal cold SST variability in the CLR is the result of combination of local and remote forcing. The remote forcing is made trough Kelvin wave eastward propagation associated with minimum z20 and SSH. For the years 2005, the May wind events was responsible to strong SST response, supported by favorable subsurface conditions. Since the subsurface conditions in the east is largely influenced by the arrival of Kelvin wave excited in the west, it seems to us interesting to

better understand what are the atmospheric conditions associated with the Kelvin wave excitation in the west and how they are different in 2005 and 2006. It is the aim of the section 4.2.2b. The main result of the analysis is that the anomalous strengthening of easterly winds occurs some days after the ITCZ to be at its southernmost location. In 2005, the ITC reaches its southernmost location through a sudden southward shift and returns to its initial position just after, whereas in 2006, the southernmost position of the ITCZ is reaches less sharply and in the continuity of the evolution of the ITCZ's position, as it is moving southward. In order to better highlight the phenomenon discussed, we have plotted the intraseasonal variations anomalies (the 30 days low-pass filtered field averaged over 1998-2008 period have been removed to the total field) of wind stress magnitude, z20, and SLA on figure 7. It also shows another way in which intraseasonal wind event impact the SST in the eastern Atlantic (even few weeks later), via the generation of Kelvin wave in the West.

**MC:** Few lines have been added at the end of the section 4.2.2b:

*"These results highlight another way in which wind intraseasonal events may impact the SST variability in the East part of the basin, through the generation of Kelvin wave in the West which shoals the thermocline in the East few weeks later"*

**RC: Lines 414-415: How does Fig. 8 show an enhancement of SST cooling after May 10? It only shows SST averaged for May and for May 1-10.**

**AC:** The enhancement of SST cooling after May 10 was deduced for the difference between the average over May and the average over May 1-10.

**MC**: For better clarity, we have modified the figure 8 and shown the mean for 1-12 May 2005 and for 14-31 May 2005. For comparison, the same calculation has been made for 2006.

[Figure]

**Figure 8**: (a) intraseasonal anomaly of sea surface temperature (° C; color) superimposed with intraseasonal anomaly of wind stress intensity (arrows) averaged over 1-12 May 2005 (up panel) and over 14-30May (down panel); (b) same but for 2006. For details about the calculations of the anomalies, see Sect.2.

The comments of the figure 8 have therefore been modified as follows:

*"To evidence the effect of these events on SST, maps of intraseasonal SST anomaly and intraseasonal wind stress anomaly averaged from 1 to 12 May (before the strong 2005 events; Fig. 8a) and from 14 to 31 May (during and after the strong 2005 events; Fig. 8b) are presented on Fig. 8. The same calculations have been for 2006 for comparison. The results illustrate an enhancement after 10 May of the cooling in the east associated with southerly wind intensification and an extension of the cooling especially south of the equator up to 20°W."*

**RC: Figure 10: Why not show anomalies for all fields instead of only for winds? It seems like sections 5.1.2 and 5.1.3 are not essential and could be eliminated.**

**AC & MC**: We modified the Figure 10 and decided to show the total field for all fields and to separate the wind pattern and the precipitation pattern for more visibility. The aim is to describe the atmospheric conditions associated to the mid-May event 2005.

[Figure]

**Figure 10**: Daily-averaged, from 13 May to 17 May 2005 (left to right panels), of (a) wind magnitude (color field) (m.s$^{-1}$) superimposed with wind vectors from CFSR fields; (b) precipitation rate (kg.m$^{-2}$/day)$^{-1}$) from CFSR fields; (b) surface pressure (hPa) from ERA-20C reanalysis; (c) wind speed curl (m.s$^{-1}$) computed from CFSR wind speed fields; and (d) downward short-wave radiation (W.m$^{-2}$) from CFSR fields.

However, we do not agree with the reviewer and do think that sections 5.1.2 and 5.1.3 are useful. Presently, the purpose of the second part of the paper is to better understand how the mid-May 2005 event is singular in addition to the anomalous strong southerly winds. These two sections show the particular conditions which accompanies the mid-May event. The section 5.1.2 shows that, through its time of occurrence and its impact on SST, the mid-May 2005 wind event has also an impact on precipitation pattern off northeast Brazil. In the section 5.1.3, we notice that the event is associated with atmospheric gravity wave which quickly propagates from south Atlantic to equatorial region, that highlights a way to carry momentum and energy of from South Atlantic region to tropical/equatorial region and raises the question of the representation of the impact of such phenomenon on the SST variability in equatorial and eastern tropical region. So, we prefer to keep these sections.

*Additional authors 'comments :*

*In addition to modifications listed above, modifications have been made to make the figures clearer and more easily understandable.*

[revised manuscript text omitted]

---

## Author Response (AR2)

Authors' response to Referee 1

*Journal* :    Ocean Sciences

*Paper title*:  Impact of intraseasonal wind bursts on SST variability in the far eastern Tropical Atlantic Ocean during boreal spring 2005 and 2006. Focus on the mid-may 2005 event.

*Authors:*    Herbert Gaëlle, Bourlès Bernard.

*We thank Reviewer 1 for his comments and suggestions.*

**RC**: Reviewer's comment; **AC**: Authors' comment; **MC**: Manuscript changes

**RC: I appreciate the authors' effort in addressing the issues raised in my review. I find the revised manuscript to be improved, but there are still a number of things that need to be taken care of before publication. In particular, I find that my major comments 1 and 2 and specific comments 1, 4, 5, 8, 10, 11, and 12 have been adequately addressed. Regarding the other comments, I still have the following concerns:**

1. **RC: A) With respect to the different time scales (related to former major comment 1 and specific comment 2 and 7), I am still having difficulty to sort out the relationship between the intraseasonal and interannual anomalies. The authors did not make it easier for me with their replies that completely mix up the time scales (AC to major comment 3 states that Fig. 4 shows interannual anomalies, the figure caption itself says intraseasonal; AC to specific comment 2 talks about interannual anomalies, but the manuscript change is about intraseasonal variations). I would strongly advise the authors to define and cleanly separate the time scales and ideally draw a conclusion how the intraseasonal episodes lead to the interannual cold event in 2005.**

**AC & MC:** Thanks for the comment**.** We apologize for the confusion. Indeed, there was a mistake in our reply to major comment 3 and to specific comment 2. We wanted to say "intraseasonal" and not "interannual". Fig.4 shows intraseasonal anomalies. The text about Fig. 3 & 4 has been modified and comments about the year 2006 have been added, as follows (section 4.1):

*"The cooling episodes occurred east of 5° E from April to September.  In 2005, the intraseasonal cooling episodes took place on  8-12 May, 16-22 May, 30 May-June 6, and 12-16 June, whereas in 2006 they took place on 20-30 April, 14-24 May, 14-20 & 26-30 June. The temperature drop for the two years ranged between -0.2°C to -2°C."*

The interannual variability is highlighted in Section 3. Then, the paper focuses on intraseasonal SST and southerly wind anomalies. This point is briefly addressed at the end of the section 4.1:

 *"Besides, model SST fields (Fig. 3a) indicate that the SST minimum (~24° C) in 2005 was reached in July, i.e. one month earlier than in 2006, as also noticed in seasonal variations of SST averaged in the region (Fig. 1a). These results illustrate the important role of the succession of quick and intense cooling episodes in the establishment of persistent cold anomalies in the CLR, as highlighted by Marin et al. (2009) in the equatorial region."*

The different processes implied in the occurrence of cooling episodes in boreal spring and the cause of differences observed from year to year are summarized in the conclusion :

*"In the CLR, stronger wind intensification and favorably preconditioned oceanic subsurface conditions in 2005 made the coupling between surface and subsurface ocean processes more efficient than in 2006, resulting in stronger cooling. It should be noted that the occurrence of intraseasonal wind intensification in the CLR is not specific to the boreal spring/summer 2005 and 2006 and is observed every year over the 1998-2008 period of study (not shown). However, their impact on SST variability in the region is modulated depending of the strength of wind intensification and of the subsurface preconditioning. For example, the year 1998, known as a "warm year", is characterized by anomalous warm SST in boreal spring/summer in the CLR., associated with anomalous weak winds and anomalous deep thermocline."*

2.**RC: B) line 196/197: The highest temperatures occur more towards boreal spring than winter (former specific comment 3). The AC states that the text has been modified, but it has not.**

**AC & MC:** Sorry for that, the text has been modified as follows:  *"The SST variations display an annual cycle with highest temperature at the end of boreal winter – beginning of boreal spring […]"*

3.**RC: C) I appreciate that Figures 3 and 4 have been revised. However, I still find it impossible to identify the individual dates of cooling episodes and wind bursts that are given in the text in those figures (former specific comment 6). Maybe you could highlight those dates by vertical lines or arrows?**

**AC:**  Black brackets and light grey lines have been added on Fig.3 in order to highlight the cooling episodes. On Fig.4, shaded areas have been added for highlight the dates of the wind bursts.
Then, the labels have been enlarged.

4.**RC: D) Regarding the seasonality of the upwelling (lines 205 to 208, 228), I believe that the passage of coastal Kelvin waves plays an important role as well (e.g. Ostrowski et al., 2009).**
**AC & MC:** Tanks for the comment. We slightly modified the text as follows:

*"The SST May-June average map indicates that the boreal summer SST minimum is related to intensified cool SST around 6°S, in the Congo mouth region. In this region, the coast is oriented parallel to the trade flow which reinforces in boreal summer, thus favorable to*

*coastal upwelling processes. The mean alongshore wind stress during May-June reveals in fact that upwelling conditions are observed over most of the CLR. The coastal upwelling could also interact with the coastal Kelvin wave propagation (e.g. Ostrowski et al,. 2009) highlighted by minimum z20 values in Fig. 1d"*

5.**RC: E) Despite some improvements, many of the figures are still very small with labels that are hard to read. I also find that the "rainbow" colormap is not ideal for showing anomalies and would advise to use a colormap centered around white or very light colors instead.**

**AC:** Thanks for the suggestion. The labels of Fig.3, 4, 5, 6, 7, 8, 9, 10, 11, 12 have been enlarged.

6.**RC: In Fig. 11b, the colormap is not very instructive either.**

**AC:** The colormap of Fig. 11b has been modified.

7.**RC: For Fig. 12, the caption is not consistent with the labels of the subplots regarding the order of wind stress and radiation.**

**AC:** The colormap of the subplot showing the radiation has been modified.

8.**RC: Figure 2 is still mentioned for the first time in the text after Figure 3.**

**AC:** Sorry for that. We have added a reference to the Fig. 2 in the Introduction, as follows:

"*The question of the processes implied in the SST variability in the Cape-Lopez region was raised based on an observation in satellite SST data of cold coastal waters during boreal spring independent from those observed off shore in the cold tongue region around 10°W (see Fig. 2) which also raised the question of the link of such cooling with the cold tongue development*."

9.**RC: F) There are also still a number of language issues. I have made some suggestions below (quite a few of them not for the first time, I believe…) but the manuscript would probably benefit from a careful read-through by a native English speaker**.
**AC:** Thanks for pointing the technical corrections. The corrections have been made.

Authors' response to Referee 2

*Journal* :   Ocean Sciences

*Paper title*:  Impact of intraseasonal wind bursts on SST variability in the far eastern Tropical Atlantic Ocean during boreal spring 2005 and 2006. Focus on the mid-may 2005 event.

*Authors:*    Herbert Gaëlle, Bourlès Bernard.

*We thank Reviewer 2 for his comments.*

**RC**: Reviewer's comment; **AC**: Authors' comment; **MC**: Manuscript changes

**RC: The authors have taken all of the reviewers' comments into account and, in my opinion, have addressed them sufficiently and made appropriate changes to the manuscript. I do still think this paper would benefit from having a native English speaker review it, though it is much better in this version. That said, the few places that still need correction do not detract from the meaning of the sentences, and the reader is able to follow along easily.**

**AC:** Thanks for your comments. Many technical corrections have been made in the revised version.

Authors' response to Referee 3

*Journal* :  Ocean Sciences

*Paper title*:  Impact of intraseasonal wind bursts on SST variability in the far eastern Tropical Atlantic Ocean during boreal spring 2005 and 2006. Focus on the mid-may 2005 event.

*Authors:*    Herbert Gaëlle, Bourlès Bernard.

*We thank Reviewer 3 for his comments and suggestions.*

**RC**: Reviewer's comment; **AC**: Authors' comment; **MC**: Manuscript changes

1.**RC: line 248: I don't see the April 22-24 cooling event in Fig. 4a. The others that you list are visible.**
**AC&MC:**  Thanks for the remark. Indeed, this is a mistake. The text has been modified and comments about the year 2006 have been also added, as follows:
*"The cooling episodes occurred east of 5° E from April to September.  In 2005, the intraseasonal cooling episodes took place on  8-12 May, 16-22 May, 30 May-June 6, and 12-16 June, whereas in 2006 they took place on 20-30 April, 14-24 May, 14-20 & 26-30 June. The temperature drop for the two years ranged between -0.2°C to -2°C."*
In addition, for more clarity, black brackets have been added on Fig.3 to indicate the dates of the cooling episodes, and shaded areas have been added in Fig.4 to indicate the southerly wind events.

2. **RC: line 289**: "**Globally" is not the right word to use here. I would simply delete it.**
**AC:** Thanks for the suggestion. It has been deleted.

3.**RC: line 340**: **You discuss the shallower thermocline being more conducive to anomalous cooling, which is probably true for vertical mixing induced by current shear and wind. However, the heat capacity (i.e., mixed layer depth) can also have a big impact on the change in SST for given heat flux, and it is not necessarily correlated with thermocline depth. I suggest checking to see if MLD follows Z20, or at least adding a disclaimer that Z20 is not necessarily the same as MLD.**
**AC & MC:**  We have slightly modified the text as follows:
*"As previously shown, the time of occurrence of the cold events in the CLR coincides with shallow thermocline which contributes to making the mixed layer temperature more reactive to surface forcing (note that z20 is not necessarily the same as the mixed layer depth)."*

4.**RC: Fig. 7**: **I suggest reversing the y axis in Fig. 7e,k so that shallow Z20 anomalies are up instead of down.**
**AC & MC:** Thanks for the suggestion. However, this would be in contradiction with another reviewer's comment. For more clarity, the line below has been added in the legend of the Fig. 7: *"Negative values indicate a 20°C isotherm depth closer to the surface"*

5.**RC: line 416**: **How does Ekman convergence reinforce upwelling? Do you mean divergence?**

**AC:** Yes, we mean divergence. This has been corrected in the text.

6. **RC: Fig. 9**: **It's a little confusing to have the longitude on the y axis (longitude goes across, not up and down like latitude). Consider putting it on the x axis and making columns instead of rows.**

**AC:** Thanks for the suggestion. The Fig. 9 has been modified.

7.**RC: line 580**: **Change 'humidify' to 'humidity'**

**AC:** Thanks. The modification has been made.

**There are numerous mostly minor language/grammatical errors that need to be corrected. It's not a major thing, but they are distracting when reading the text.**
Many corrections have been made.
In addition, the labels of most of the figures have been enlarged.

[revised manuscript text omitted]

---

## Author Response (AR3)

Authors' response to Editor's comments

*Journal* :             Ocean Sciences

*Title of the paper*:  Impact of intraseasonal wind bursts on sea surface temperature variability in the far eastern Tropical Atlantic Ocean during boreal spring 2005 and 2006. Focus on the mid-May 2005 event.

*Authors*:             Herbert Gaëlle, Bourlès Bernard

**EC:** Editor 's comments; **AC:** Authors' comments

**EC: You have satisfactorily addressed the comments by the referee and your manuscript is now accepted for publication in Ocean Science. Before submitting the final version of your manuscript, please account for the last few technical issues listed below.**

**AC:** We would like to thank the editor for his careful reading and valuable remarks. Additional small changes have been made following them as well as those offered by the reviewer, and we hope the revised manuscript will better suit the *Ocean Science*.

**EC: Title, I suggest a minor modification (no abbreviations in the title; colon after the first part): Impact of intraseasonal wind bursts on sea surface temperature variability in the far eastern tropical Atlantic Ocean during boreal spring 2005 and 2006: Focus on the mid-May 2005 event**
**AC:** The modification has been made.

**EC:** **-L20, L21 anomalously strong**
**-L21 … triggered the onset of coastal rainfall in the … (the text was confusing; is this correct?)**
**-L22-23 I suggest: No similar atmospheric conditions were observed in May over the 1998-2008 period.**
**-L25 from the South Atlantic**
**-L147 "theta_s = 6 and theta_b = 0" Is this correct (format)?**
**-Data used: Is there any way to quantify the precision of the variables used?**
**You use data from other sources. Please check the fair data use policies and whether**

**you acknowledge the data originators adequately.**

**-L329, L339 1 March (format)**

**-L415 10 Feb 2006**

**-L553-554 please use consistent format**

**-L703 McPhaden**

**-L755, L758 Foltz et al., 2006 should be Foltz et al., 2003 according to the reference list**

**-L879 Journal: Science (typo), and change format.**

**-L1018 Deep-Sea Res.**

**AC:** Thank you for pick up technical issues. The corrections have been made.

About "theta_s" and "theta-b", it is the right format.

About the data used, we modified the text as follows for TMI data:

*"The dataset is a merged product produced by Remote Sensing Systems and sponsored by the NASA Earth Sciences Program. The data are available at www.remss.com/missions/tmi."*
*(l.170-171).*

For sea surface height dataset:

*"We also use for this study daily sea surface height (SSH) data, which are available for the period 1993–2012 and maintained by the organization for Archiving, Validation, and Interpretation of Satellite Oceanographic data with support from Cnes (AVISO; www.aviso.altimetry.fr/duacs). The sea surface height dataset is a merged product of observations from several satellite missions Ssalto/Duacs (Segment Sol multimissions d'ALTimétrie, d'Orbitographie et de localisation précise/Developing Use of Altimetry for Climate Studies) mapped onto a 0.25° Mercator projection grid."*

For ECMWF fields:

*"In addition, surface pressure data were studied using ECMWF Atmospheric Reanalysis (ERA) for the 20th Century product (European Centre for Medium-Range Weather Forecasts, 2014). The four-hourly data are daily averaged and is available on https://rda.ucar.edu website. The product assimilates surface pressure and marine wind observations."*

About "Foltz et al., 2006" (l.755, 758), it is the correct citation. It refers to the reference noted in the reference list:

*Foltz, G.R. and McPhaden, M.J.: Unusually warm sea surface temperatures in the tropical*

*North Atlantic during 2005, Geophys. Res. Lett., 33: doi: 10.1029/2006GL027394. issn:*

*0094-8276, 2006.*

**Authors' response to reviewer's comments**

*Journal* :  Ocean Sciences

*Title of the paper*:  Impact of intraseasonal wind bursts on sea surface temperature variability in the far eastern Tropical Atlantic Ocean during boreal spring 2005 and 2006. Focus on the mid-May 2005 event.

*Authors*:  Herbert Gaëlle, Bourlès Bernard

**RC:** Reviewer 's comments; **AC:** Authors' comments

**RC: The authors have adequately addressed most of my concerns with the previous version of their manuscript. I recommend publication with some minor additional changes listed below.**

**AC:** We would like to thank again the reviewer for the time and effort necessary to provide constructive remarks that contributed to improving the final version of our manuscript.

**RC:** Reviewer's comments; **AC:** Authors' comments

**RC:**
**line 12: wind anomalies**
**line 165 and 167: removing FROM (not to)**
**line 256: Do you mean zonal?**
**line 346/347: z20 should most definitely not be the same as the mixed layer. I think what you mean to say is that mixed layer depth and thermocline depth do not necessarily show the same variability.**
**line 625, 628: RADSW needs to be explained (short wave radiation, I presume)**
**line 630: lowest (instead of weakest)**
**line 698/699: I find it very hard to follow that sentence.**
**line 729: as pronounced AS**
**line 733: I would suggest "importance" (instead of impact)**
**For Fig. 12, the caption is still not consistent with the labels of the subplots regarding the order of wind stress and radiation.**

**AC:** Thanks for technical corrections. The modifications have been made.
About RADSW (line 625, 628 of the old version), the term is explained few lines above (line 618, 619 of the old version, line 620, 621 of the new version).

[revised manuscript text omitted]